# Are High-Degree Representations Really Unnecessary in Equivariant Graph Neural Networks?

**Jiacheng Cen**[1 2], **Anyi Li**[1 2], **Ning Lin**[1 2], **Yuxiang Ren**[3], **Zihe Wang**[1 2], **Wenbing Huang**[1 2*]

[1] Gaoling School of Artificial Intelligence, Renmin University of China
[2] Beijing Key Laboratory of Big Data Management and Analysis Methods
[3] 2012 Laboratories, Huawei Technologies, Shanghai
{jiacc.cn, li_anyi, ninglin00}@outlook.com; renyuxiang1@huawei.com;
wang.zihe@ruc.edu.cn; hwenbing@126.com

## Abstract

Equivariant Graph Neural Networks (GNNs) that incorporate the E(3) symmetry have achieved significant success in various scientific applications. As one of the most successful models, EGNN [1] leverages a simple scalarization technique to perform equivariant message passing over only Cartesian vectors (i.e., 1st-degree steerable vectors), enjoying greater efficiency and efficacy compared to equivariant GNNs using higher-degree steerable vectors. This success suggests that higher-degree representations might be unnecessary. In this paper, we disprove this hypothesis by exploring the expressivity of equivariant GNNs on symmetric structures, including $k$-fold rotations and regular polyhedra. We theoretically demonstrate that equivariant GNNs will always degenerate to a zero function if the degree of the output representations is fixed to 1 or other specific values. Based on this theoretical insight, we propose HEGNN, a high-degree version of EGNN to increase the expressivity by incorporating high-degree steerable vectors while still maintaining EGNN's advantage through the scalarization trick. Our extensive experiments demonstrate that HEGNN not only aligns with our theoretical analyses on a toy dataset consisting of symmetric structures, but also shows substantial improvements on other complicated datasets without obvious symmetry, including $N$-body and MD17. Our study potentially showcase an effective way of modeling high-degree representations in equivariant GNNs.

## 1 Introduction

Molecules, proteins, crystals, and many other scientific data can be effectively modeled and represented through *geometric graphs* [2–8]. This type of data structure encapsulates not only node characteristics and edge information but also a 3D vector (such as position, velocity, etc.) for each node. To process geometric graphs, equivariant Graph Neural Networks (GNNs) have been developed, which undergo equivariant message passing over nodes, conforming to the E(3) or SE(3) symmetry of physical laws. These models have achieved remarkable successes in a lot of scientific tasks, such as physical dynamics simulation [9–11], molecular generation [12–15] and protein design [16–18].

Pioneer equivariant GNNs [19–22] derive high-degree steerable representations beyond scalars and 3D coordinates with the help of spherical harmonics and conduct equivariant message passing between representations of different degrees through the Clebsch-Gordan (CG) tensor product. While these high-degree models are able to approximate any function of fully connected geometric graphs in theory [23], they usually suffer from expensive computational costs in practice. In contrast,

---

*Wenbing Huang is the corresponding author.

38th Conference on Neural Information Processing Systems (NeurIPS 2024).

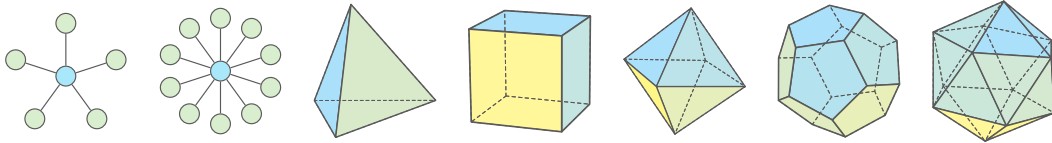

$k$-fold (odd)    $k$-fold (even)    Tetrahedron    Cube (Hexahedron)    Octahedron    Dodecahedron    Icosahedron

Figure 1: Common symmetric graphs. Equivariant GNNs on symmetric graphs will degenerate to a zero function if the degree of their representations is fixed as 1.

EGNN [1] leverages a simple scalarization technique to allow equivariant message passing over only 3D vectors (*i.e.* the 1st-degree steerable features). Specifically, the scalarization technique first encodes 3D vectors into scalars as invariant messages, which are passed as geometric messages after the multiplication with the 3D vectors to recover the orientation information. Despite its simplicity, EGNN achieves remarkably better efficacy and efficiency against conventional high-degree models for a broad range of applications [24; 25]. Such successes suggest that higher-degree representations might be unnecessary.

In this paper, we challenge and disprove this hypothesis by exploring the expressivity of equivariant GNNs on symmetric graphs. Fig. 1 illustrates the examples of $k$-fold rotations and regular polyhedra, which are invariant to rotations up to certain rotating angles. Taking the cube for example, conducting $90°$ rotation around the axes crossing the center of the two opposite faces keeps its shape and orientation unchanged. Interestingly, by making use of group theory, we theoretically prove that any equivariant GNN (after translating the coordinate center to the origin and conducting graph-level readout) on these symmetric graphs will degenerate to a zero function if the degree of their representations is fixed to be 1. The direct deduction of this theorem is that EGNN can only output a zero 3D vector no matter how we rotate the input graph, indicating that EGNN totally loses the recognition ability of orientation. Additionally, this statement points out the limitation of the methods that rely on constructing global features for symmetric graphs [2] (*e.g.* frames in frame averaging [26; 27], virtual nodes in FastEGNN [28]), equivariant pooling in EGHN [29], and meshes in Neural P$^3$M [30]), since it is impossible to output another non-collinear 3D vector except the center coordinate.

Based on the above theoretical insights, we propose a novel equivariant GNN model termed HEGNN[3], which enhances EGNN by incorporating high-degree steerable vectors while inheriting the desired benefit from EGNN through the scalarization trick. In summary, our contributions are as follows:

- We theoretically investigate the expressivity reduction issue of equivariant GNNs on symmetric graphs.

- We propose HEGNN, to further incorporate high-degree steerable representations into EGNN. Moreover, since the equivariant message passing process between different-degree representations is conducted via inner products, it shares the same benefit as EGNN, compared to traditional high-degree models.

- Our extensive experiments demonstrate that HEGNN not only aligns with our theoretical analyses on toy datasets consisting of symmetric graphs, but also shows substantial improvements on more complicated datasets without explicit symmetry, such as $N$-body and MD17.

## 2   Related Works

**Equivariant GNNs.** Equivariant GNNs can be divided into two classes: scalarization-based models and high-degree steerable models [31]. Scalarization-based models adopt norms or inner products to convert equivariant 3D vectors into invariant scalars, which are considered as coefficients to linearly combine 3D vectors for node update. EGNN [1] is the first work falling into this category. Concurrently, PAINN [32] further enhances the expressive ability of the model by introducing multi-channel equivariant features. On the contrary, high-degree steerable models (*e.g.* TFN [19],

---

[2]See Appendix A.3 for further discussion.
[3]Code is available at https://github.com/GLAD-RUC/HEGNN.

SEGNN [33] and SE(3)-Transformer) use spherical harmonics to ensure the equivariance of message passing, and realize interaction between steerable features of different degrees through CG tensor products. Our HEGNN also uses high-degree steerable features, but it leverages the scalarization trick for the interaction between steerable features of different degrees, thus leading to more expressivity than EGNN and less computational cost than other high-degree models.

**Expressivity of Equivariant GNNs.** The theoretical expressivity of equivariant GNNs is initially explored by [23], which proves the university of the high-degree steerable model, *i.e.*, TFN [19], over fully-connected geometric graphs. GemNet [34] further demonstrates that the universality holds with just spherical representations other than the full $\mathrm{SO}(3)$ representations that are required in the proof of [23]. More recently, the GWL framework [35] extends the Weisfeiler-Lehman (WL) test into a geometric version [36] to study the expressive power of geometric GNNs operating on sparse graphs from the perspective of discriminating geometric graphs. Different from all the above works, our paper investigates the expressivity of equivariant GNNs on symmetric graphs, and demonstrates the necessity of involving high-degree representations. Although the GWL test paper [35] has experimentally compared different models on $k$-fold structures that are allowed to rotate only in the 2D space, the conclusions of this paper are proved both theoretically and experimentally. Moreover, our discussions cover a full range of examples including $k$-folds (rotation in 3D space) and regular polyhedra.

## 3 Theoretical Analyses

In this section, we first present the necessary preliminaries related to geometric graphs and group representation. Then, we define and illustrate typical examples of symmetric graphs. Finally, we will discuss when equivariant GNNs will degenerate to a zero function on symmetric graphs.

### 3.1 Preliminaries

**Geometric graph.** A geometric graph of $N$ nodes is defined as $\mathcal{G} := \left(\boldsymbol{H}, \vec{\boldsymbol{X}}; \boldsymbol{A}\right)$, where $\boldsymbol{H} := \{\boldsymbol{h}_i \in \mathbb{R}^{C_H}\}_{i=1}^{N}$ and $\vec{\boldsymbol{X}} := \{\vec{\boldsymbol{x}}_i \in \mathbb{R}^3\}_{i=1}^{N}$ are node features and 3D coordinates, respectively; $\boldsymbol{A} \in \mathbb{R}^{N \times N}$ represents the adjacency matrix and can be assigned with edge features $\boldsymbol{e}_{ij}$ if necessary. Throughout our theoretical analyses in this section, we assume the node features and edge features to be identical for all.

**Transformation of geometric graph.** We are interested in the transformations of a geometric graph $\mathcal{G}$ with respect to a group $\mathfrak{G}$, which is defined as $\mathfrak{g} \cdot \mathcal{G}$, for $\mathfrak{g} \in \mathfrak{G}$ and $\cdot$ denoting the group action. For instance, $\mathfrak{g} \cdot \mathcal{G}$ can be explained as translation, rotation, or reflection of the coordinates $\vec{\boldsymbol{X}}$. These transformations form a 3D Euclidean group denoted as $\mathrm{E}(3)$, and its subgroup without translation is called the orthogonality group $\mathrm{O}(3)$. With the aid of group representation $\rho(\mathfrak{g})$, the transformation of a coordinate $\vec{\boldsymbol{x}}$ is represented as $\rho(\mathfrak{g})\vec{\boldsymbol{x}}$. For example, orthogonal matrices are the trivial representations of $\mathrm{O}(3)$, that is, the orthogonal transformation of a vector $\vec{\boldsymbol{x}}$ is represented by $\boldsymbol{O}\vec{\boldsymbol{x}}$ with $\boldsymbol{O} \in \mathbb{R}^{3 \times 3}$ being an orthogonal matrix. Besides, there are other representations of $\mathrm{O}(3)$, such as the irreducible representations which will be detailed below.

**Equivariance.** Let $\mathcal{X}$ and $\mathcal{Y}$ be the input and output vector spaces, respectively. A function $f : \mathcal{X} \to \mathcal{Y}$ is called *equivariant* with respect to group $\mathfrak{G}$ if

$$\forall \mathfrak{g} \in \mathfrak{G}, f(\rho_{\mathcal{X}}(\mathfrak{g})\vec{\boldsymbol{x}}) = \rho_{\mathcal{Y}}(\mathfrak{g})f(\vec{\boldsymbol{x}}), \tag{1}$$

where $\rho_{\mathcal{X}}$ and $\rho_{\mathcal{Y}}$ are the group representations in the input and output spaces, respectively. Since we can eliminate the translation effect by simply translating the center of all coordinates to the origin, we only discuss equivariance with respect to $\mathrm{O}(3)$ in this section. In other words, we default that the center of $\vec{\boldsymbol{X}}$ is at the origin.

**Irreducible representations and steerable features.** $\mathrm{O}(3)$ consists of rotation and inversion, implying $\mathrm{O}(3) = \mathrm{SO}(3) \times C_i$, where $\mathrm{SO}(3)$ is the rotation group and $C_i = \{\mathfrak{e}, \mathfrak{i}\}$ denotes the inverse group. We first discuss the irreducible representations of $\mathrm{SO}(3)$. For each rotation $\mathfrak{r} \in \mathrm{SO}(3)$, its irreducible representations are Wigner-D matrices $\boldsymbol{D}^{(l)}(\mathfrak{r}) \in \mathbb{R}^{(2l+1) \times (2l+1)}$ of different degree $l \in \mathbb{N}$ [21; 37]. When $l = 1$, it becomes the common rotation matrix $\boldsymbol{R}_{\mathfrak{r}}$ acting on the 3D coordinate space. Under the irreducible representations, the equivariant constraint in Eq. (1) turns

into $f^{(l)}(\boldsymbol{R}_\mathfrak{r}\vec{\boldsymbol{x}}) = \boldsymbol{D}^{(l)}(\mathfrak{r})f^{(l)}(\vec{\boldsymbol{x}})$, if the output degree is $l$. According to [38], spherical harmonics $Y^{(l)} = [Y_m^{(l)}(\vec{\boldsymbol{x}})]_{m=-l}^l$ offer a *unique* and *complete* set of function bases satisfying the equivariant constraint. We further define a modulated spherical harmonics as $f^{(l)}(\vec{\boldsymbol{x}}) = \varphi(\|\vec{\boldsymbol{x}}\|) \cdot Y^{(l)}(\vec{\boldsymbol{x}}/\|\vec{\boldsymbol{x}}\|)$ by adding a continuous radial function $\varphi : \mathbb{R}^+ \to \mathbb{R}$ of vector norm $\|\cdot\|$ for re-scaling. Such a function $f^l$ and its output $f^l(\vec{\boldsymbol{x}})$ are called type-$l$ *steerable function* and *steerable feature*, respectively. We now deduce the irreducible representations from $\mathrm{SO}(3)$ to $\mathrm{O}(3)$. Note that spherical harmonics satisfy $Y^{(l)}(-\vec{\boldsymbol{x}}) = (-1)^l Y^{(l)}(\vec{\boldsymbol{x}})$; in other words, they are inverse-equivariant when $l$ is odd, but inverse-invariant when $l$ is even. We thus specify the group representation of $\mathrm{O}(3)$ as

$$\rho^{(l)}(\mathfrak{rm}) := \sigma^{(l)}(\mathfrak{m})\boldsymbol{D}^{(l)}(\mathfrak{r}), \tag{2}$$

where $\sigma^{(l)}(\mathfrak{m}) = 1$ for $\mathfrak{m} = \mathfrak{e}$ (the identity) and $\sigma^{(l)}(\mathfrak{m}) = (-1)^l$ if $\mathfrak{m} = \mathfrak{i}$ (the inverse). Readers can refer to the discussion in e3nn [39] with another representation method by using the concept of *parity* and construct this through methods such as Clebsch-Gordan (CG) tensor product [40]. For concision, the type-$l$ steerable feature is denoted as $\tilde{\boldsymbol{v}}^{(l)}$ with a tilde notation.

## 3.2 Symmetric Graph

In § 1, we present that $k$-fold rotations and regular polyhedra exhibit certain symmetries. In this subsection, we formally describe them via the notion of the symmetric graph.

**Definition 3.1** (Symmetric Graph). A geometric graph $\mathcal{G}$ is called a symmetric graph, if there exists a finite and nontrivial subgroup $\mathfrak{H} \leq \mathrm{O}(3), \mathfrak{H} \neq \{\mathfrak{e}\}$, satisfying that $\forall \mathfrak{h} \in \mathfrak{H}, \mathfrak{h} \cdot \mathcal{G} = \mathcal{G}$. All subgroups making $\mathcal{G}$ symmetric yields a set $\mathbb{H}(\mathcal{G})$, and all geometric graphs that are symmetric *w.r.t.* $\mathfrak{H}$ constitute a set denoted as $\mathbb{G}(\mathfrak{H})$.

Here $\mathfrak{h} \cdot \mathcal{G} = \mathcal{G}$ is defined in the graph level. Particularly for the coordinates $\vec{\boldsymbol{X}} \in \mathbb{R}^{3 \times N}$, it implies that $\forall \boldsymbol{O} \in \mathfrak{H}, \exists \boldsymbol{P} \in S_N, \boldsymbol{O}\vec{\boldsymbol{X}} = \vec{\boldsymbol{X}}\boldsymbol{P}$ and $\boldsymbol{P}\boldsymbol{A} = \boldsymbol{A}\boldsymbol{P}$, where $S_N$ is the permutation group of order $N$. Essentially, rotating the coordinates of a symmetric graph leads to a copy of this graph up to a different permutation of the nodes.

Without considering inversion, the finite subgroups of $\mathrm{SO}(3)$ are only cyclic group $C_n$, dihedral group $D_n$, tetrahedral group $T$, octahedral group $O$, and Icosahedral group $I$ [41]. We provide several examples of symmetric graphs as follows.

**Example 3.2** ($k$-folds). On the one hand, for a geometric graph $\mathcal{G}$ corresponding to a $2k$-fold with nodes $\{(\cos(i \cdot \pi/k), \sin(i \cdot \pi/k), 0)\}_{i=0}^{2k-1}$, the inverse group $C_i$ and the dihedral group $D_{2k}$ (rotation around $z$-axis with angle $\pi/k$, and reflection around the axis connecting the midpoints of opposite sides or the axis connecting opposite vertices), are symmetric groups on $\mathcal{G}$, namely, $C_i, D_{2k} \in \mathbb{H}(\mathcal{G})$. On the other hand, for a geometric graph $\mathcal{G}$ corresponding to a $(2k+1)$-fold with nodes $\{(\cos(i \cdot 2\pi/(2k+1)), \sin(i \cdot 2\pi/(2k+1)), 0)\}_{i=0}^{2k}$, $\mathbb{H}(\mathcal{G})$ includes the dihedral group $D_{2k+1}$ but without the inverse group $C_i$.

**Example 3.3** (Regular Polygons). The symmetric groups of regular polygons in the plane and regular prisms in space include the dihedral group $D_n$. Regular tetrahedra are symmetric with respect to three rotation axes of the second order and four axes of the third order, corresponding to 12 group elements. Regular hexahedra (cubes) and the regular octahedra (which are dual to each other and share the same symmetric groups) are symmetric about six axes of the second order, four axes of the third order, and three axes of the fourth order, corresponding to 24 group elements. Regular dodecahedra and regular icosahedra (which are also dual to each other) are symmetric about six axes of the fifth order, ten axes of the third order, and fifteen axes of the second order, corresponding to 60 group elements. Additionally, except tetrahedra, all other four regular polygons are central symmetric, indicating that $C_i$ is their common symmetric group.

## 3.3 Equivariant GNNs on symmetric graphs

We now demonstrate that equivariant GNNs on symmetric graphs will encounter the issue of expressivity degeneration. Here, we assume that the graph functions we explore are invariant to the permutation of the nodes. This fits the case when we add a readout layer to all nodes globally or just focus on the message passing process for each node individually.

We first derive a crucial theorem that greatly facilitates our analyses.

**Theorem 3.4.** *Suppose that $f^{(l)}$ is an $\mathrm{O}(3)$-equivariant function on geometric graphs, regarding the group representation $\rho^{(l)}$ defined in Eq. (2). Then, for any symmetric graph $\mathcal{G}$ induced by the group $\mathfrak{H} \leq \mathrm{O}(3)$, namely, $\forall \mathcal{G} \in \mathbb{G}(\mathfrak{H})$, we always have*

$$f^{(l)}(\mathcal{G}) = \rho^{(l)}(\mathfrak{H}) f^{(l)}(\mathcal{G}). \tag{3}$$

*Here we have defined group average as $\rho^{(l)}(\mathfrak{H}) := \frac{1}{|\mathfrak{H}|} \sum_{\mathfrak{h} \in \mathfrak{H}} \rho^{(l)}(\mathfrak{h})$.*

Eq. (3) is interesting and it shows that the function $f^{(l)}$ is symmetric with respect to the group average $\rho^{(l)}(\mathfrak{H})$. More importantly, it indicates an linear equation $\left(\boldsymbol{I}_{2l+1} - \rho^{(l)}(\mathfrak{H})\right) f^{(l)}(\mathcal{G}) = 0$, where $\boldsymbol{I}_{2l+1} \in \mathbb{R}^{(2l+1) \times (2l+1)}$ is the identity matrix. We can immediately attain the following conclusion.

**Theorem 3.5.** *If and only if the matrix $\boldsymbol{I}_{2l+1} - \rho^{(l)}(\mathfrak{H})$ is non-singular, the $\mathrm{O}(3)$-equivariant function $f^{(l)}$ is always a zero function on $\mathcal{G}$, namely,*

$$f^{(l)}(\mathcal{G}) \equiv \boldsymbol{0}, \quad \forall \mathcal{G} \in \mathbb{G}(\mathfrak{H}). \tag{4}$$

A more general version of Theorem 3.5 is that the output space of $f^{(l)}$ corresponds to the null space of the matrix $\boldsymbol{I}_{2l+1} - \rho^{(l)}(\mathfrak{H})$, indicating that $\dim(f^{(l)}) = (2l+1) - \mathrm{rank}\left(\boldsymbol{I}_{2l+1} - \rho^{(l)}(\mathfrak{H})\right)$. Therefore, even the function $f^{(l)})$ will not exactly reduce to a zero function when $\boldsymbol{I}_{2l+1} - \rho^{(l)}(\mathfrak{H})$ is singular, its output space is still limited to a subspace and suffers from diminished expressivity owing to the symmetry of the input geometric graph.

In practice, it is difficult to determine if the matrix $\boldsymbol{I}_{2l+1} - \rho^{(l)}(\mathfrak{H})$ is singular. This determination becomes easier if we can show that the group average $\rho^{(l)}(\mathfrak{H})$ is equal to the zero matrix. Fortunately, we have the following property.

**Theorem 3.6.** *For a finite group $\mathfrak{H}$ with its representation $\rho^{(l)}$, $\rho^{(l)}(\mathfrak{H})$ is a zero matrix (i.e., $\rho^{(l)}(\mathfrak{H}) = \boldsymbol{0}$) if and only if $\mathrm{tr}(\rho^{(l)}(\mathfrak{H})) = 0$. In this case, $f^{(l)}(\mathcal{G}) \equiv \boldsymbol{0}, \forall \mathcal{G} \in \mathbb{G}(\mathfrak{H})$.*

According to Theorem 3.6, we calculate the trace of the group average for each symmetric graph of interest and check if the trace is equal to zero. We summarize the conclusions for $k$-fold structures and regular polyhedra in Table 1. We find that when $l = 1$, $f^{(1)} \equiv \boldsymbol{0}$ for all cases. In addition, the function degenerates when $l$ is odd, if the symmetric graph is induced by the inverse group $C_i$. We defer more details of the calculations in the Appendix. Compared to the conclusions drawn by the GWL paper [35] which only experimentally discusses the $k$-fold structures under 2D rotations, here we apply rigorous theoretical derivations to analyze more cases besides $k$-folds, regarding more symmetric subgroups of $\mathrm{O}(3)$.

Table 1: Expressivity degeneration of equivariant GNNs on symmetric graphs.

| Symmetric Graph $\mathcal{G}$ | Symmetry Group $\mathfrak{H} \in \mathbb{H}(\mathcal{G})$ | $l$ leading to $f^{(l)}(\mathcal{G}) \equiv \boldsymbol{0}$ |
| --- | --- | --- |
| $2k$-fold | $C_i, D_{2k}$ | $l$ is odd |
| $(2k+1)$-fold | $D_{2k+1}$ | $l < 2k+1$ and $l$ is odd |
| Tetrahedron | $T$ | $l \in \{1, 2, 5\}$ |
| Cube/Octahedron | $C_i, O$ | $l = 2$ or $l$ is odd |
| Dodecahedron/Icosahedron | $C_i, I$ | $l \in \{2, 4, 8, 14\}$ or $l$ is odd |

## 4 The Proposed HEGNN

The analyses in the last section imply the necessity of involving the representations with more and higher degrees in equivariant GNNs. Therefore, we propose HEGNN by further conducting the update of high-degree steerable features upon EGNN [1]. As illustrated in Fig. 2, HEGNN is composed of the three key components: initialization of high-degree steerable features, calculation of cross-degree invariant messages, and aggregation of neighbor messages, the latter two of which are conducted over multiple layers. We depict each component separately in what follows.

**Initialization of high-degree steerable features.** Given a geometric graph $\mathcal{G}\left(\boldsymbol{H}, \vec{\boldsymbol{X}}; \boldsymbol{A}\right)$ where each node contains only type-0 feature $\boldsymbol{h}_i$ and type-1 feature $\vec{\boldsymbol{x}}_i$, we first obtain the initialization

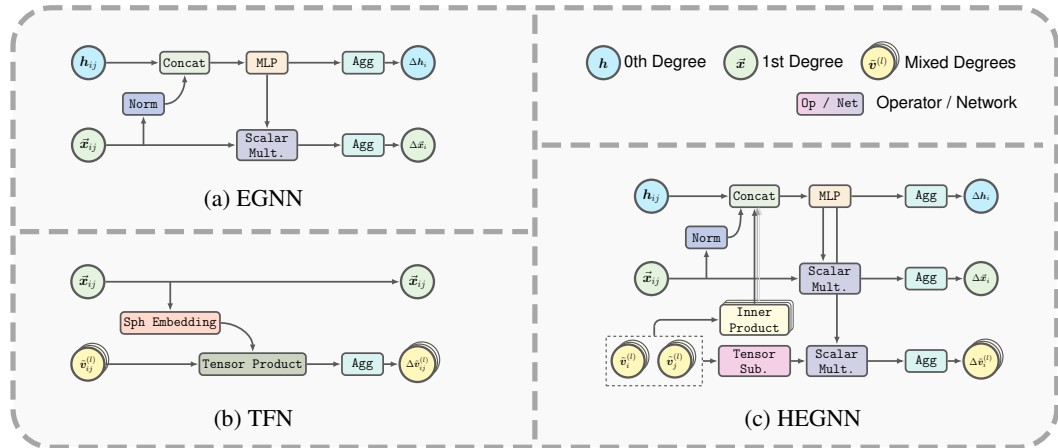

Figure 2: The different architectures of our HEGNN, EGNN [1] and TFN [19]. HEGNN exploits the scalarization trick inspired by EGNN to enable steerable features to interact between different degrees, avoiding the high computational cost of using CG tensor products in TFN.

of high-degree steerable features $\{\tilde{\boldsymbol{v}}_i^{(l)}\}_{l=0}^{L}$ by using spherical harmonics on normalized relative coordinates. In detail, we aggregate spherical harmonics from all neighbors as

$$\tilde{\boldsymbol{v}}_{i,\text{init}}^{(l)} = \frac{1}{|\mathcal{N}(i)|} \sum_{j \in \mathcal{N}(i)} \varphi_{\tilde{\boldsymbol{v}},\text{init}}^{(l)}(\boldsymbol{m}_{ij,\text{init}}) \cdot Y^{(l)}\left(\frac{\vec{\boldsymbol{x}}_i - \vec{\boldsymbol{x}}_j}{\|\vec{\boldsymbol{x}}_i - \vec{\boldsymbol{x}}_j\|}\right), \tag{5}$$

where $\boldsymbol{m}_{ij,\text{init}} = \varphi_{\boldsymbol{m},\text{init}}\left(\boldsymbol{h}_i, \boldsymbol{h}_j, \boldsymbol{e}_{ij}, d_{ij}^2\right)$ is an invariant scalar with $\varphi_{\boldsymbol{m},\text{init}}$ being an arbitrary MultiLayer Perceptron (MLP), and $\mathcal{N}(i)$ denotes the neighbors of $i$[4].

**Calculation of cross-degree invariant messages.** EGNN [1] employs a scalarization trick by transforming the relative coordinate $\vec{\boldsymbol{x}}_i - \vec{\boldsymbol{x}}_j$ (the usage of relative coordinates is for translation invariance) into an invariant scalar via the vector norm, which will be used to compute invariant message for both node features and coordinates. We generalize this scalarization trick to the case of high-degree steerable features. To be specific, we carry out the inner product between $\tilde{\boldsymbol{v}}_i^{(l)}$ and $\tilde{\boldsymbol{v}}_j^{(l)}$ for each degree $l$ individually, resulting in an invariant scalar $z_{ij}^{(l)}$. Then, we get the invariant message between node $i$ and $j$, namely $\boldsymbol{m}_{ij}$ after undergoing an MLP of all invariant quantities. The above processes are summarized as follows:

$$d_{ij} = \|\vec{\boldsymbol{x}}_i - \vec{\boldsymbol{x}}_j\|, \quad z_{ij}^{(l)} = \left\langle \tilde{\boldsymbol{v}}_i^{(l)}, \tilde{\boldsymbol{v}}_j^{(l)} \right\rangle, \quad \boldsymbol{m}_{ij} = \varphi_{\boldsymbol{m}}\left(\boldsymbol{h}_i, \boldsymbol{h}_j, \boldsymbol{e}_{ij}, d_{ij}^2, \bigoplus_{l=0}^{L} z_{ij}^{(l)}\right), \tag{6}$$

where $\bigoplus$ refers to concatenation. It should be noted that the form of SO3KRATES introduced in the concurrent work [42] is equivalent to the expression for $z_{ij}^{(l)}$ in Eq. (6). Furthermore, our scalarization trick simplifies the formulation by bypassing the Clebsch-Gordan coefficients, making it more straightforward and easier to understand.

**Aggregation of neighbor messages.** With the invariant message $\boldsymbol{m}_{ij}$ at hand, we then update $\boldsymbol{h}_i, \vec{\boldsymbol{x}}_i, \tilde{\boldsymbol{v}}_i^{(l)}$ via message aggregation over all neighbors. We define $\Delta \boldsymbol{h}_i, \Delta \vec{\boldsymbol{x}}_i, \Delta \tilde{\boldsymbol{v}}_i^{(l)}$ as the residues, which are calculated by:

$$\Delta \boldsymbol{h}_i = \varphi_{\boldsymbol{h}}\left(\boldsymbol{h}_i, \frac{1}{|\mathcal{N}(i)|} \sum_{j \in \mathcal{N}(i)} \boldsymbol{m}_{ij}\right), \quad \Delta \vec{\boldsymbol{x}}_i = \frac{1}{|\mathcal{N}(i)|} \sum_{j \in \mathcal{N}(i)} \varphi_{\vec{\boldsymbol{x}}}(\boldsymbol{m}_{ij}) \cdot (\vec{\boldsymbol{x}}_i - \vec{\boldsymbol{x}}_j), \tag{7}$$

$$\Delta \tilde{\boldsymbol{v}}_i^{(l)} = \frac{1}{|\mathcal{N}(i)|} \sum_{j \in \mathcal{N}(i)} \varphi_{\tilde{\boldsymbol{v}}}^{(l)}(\boldsymbol{m}_{ij}) \cdot \left(\tilde{\boldsymbol{v}}_i^{(l)} - \tilde{\boldsymbol{v}}_j^{(l)}\right), \tag{8}$$

---

[4]Eq. (5) is unable to derive pseudo-vectors such as torque or angular momentum, which are type-1 steerable features but invariant to reflection. To address this issue, we can further conduct CG tensor product between $\tilde{\boldsymbol{v}}_{i,\text{init}}^{(l)}$ and $\vec{\boldsymbol{x}}_i - \vec{\boldsymbol{x}}_j$ to yield the steerable feature of desired symmetry.

where $\varphi_{\boldsymbol{h}}, \varphi_{\vec{\boldsymbol{x}}}, \varphi_{\tilde{\boldsymbol{v}}}^{(l)}$ are different MLPs, and $\varphi_{\vec{\boldsymbol{x}}}, \varphi_{\tilde{\boldsymbol{v}}}^{(l)}$ both output a 1D scalar. Note that the application of Eq. (8) for all degrees can be compactly rewritten as $\bigoplus_{l=0}^{L} \Delta \tilde{\boldsymbol{v}}_i^{(l)} = \frac{1}{|\mathcal{N}(i)|} \sum_{j \in \mathcal{N}(i)} 1 \otimes_{\text{cg}}^{\varphi_{\tilde{\boldsymbol{v}}}(\boldsymbol{m}_{ij})} \left( \bigoplus_{l=0}^{L} \left( \tilde{\boldsymbol{v}}_i^{(l)} - \tilde{\boldsymbol{v}}_j^{(l)} \right) \right)$ in the form of CG tensor product with the weights $\varphi_{\tilde{\boldsymbol{v}}}(\boldsymbol{m}_{ij}) := \bigoplus_{l=0}^{L} \varphi_{\tilde{\boldsymbol{v}}}^{(l)}$. This form can be easily implemented using existing libraries such as `e3nn.o3.FullyConnectedTensorProduct` [39]. The resulting residues are used for the update:

$$\boldsymbol{h}_i = \boldsymbol{h}_i + \Delta \boldsymbol{h}_i, \quad \vec{\boldsymbol{x}}_i = \vec{\boldsymbol{x}}_i + \Delta \vec{\boldsymbol{x}}_i, \quad \tilde{\boldsymbol{v}}_i^{(l)} = \tilde{\boldsymbol{v}}_i^{(l)} + \Delta \tilde{\boldsymbol{v}}_i^{(l)}. \tag{9}$$

In addition, we can augment the update of $\vec{\boldsymbol{x}}_i$ with 1st-degree feature $\tilde{\boldsymbol{v}}_i^{(1)}$, leading to $\vec{\boldsymbol{x}}_i = \vec{\boldsymbol{x}}_i + \Delta \vec{\boldsymbol{x}}_i + \phi_{\tilde{\boldsymbol{v}}}^{(1)}(\boldsymbol{h}_i) \tilde{\boldsymbol{v}}_i^{(1)}$, which yet is not explored in our experiments for the sake of simplicity. The final output of $\boldsymbol{h}_i$ and $\vec{\boldsymbol{x}}_i$ can be used for the node-level invariant prediction and equivariant prediction, respectively. We can also obtain a graph-level prediction by further adding a readout layer of all nodes.

We now analyze the expressivity of HEGNN. Apparently, by including high-degree features, HEGNN is able to avoid the loss of expressive ability even on symmetric graphs. Moreover, when tackling general geometric graphs, HEGNN is capable of characterizing the complete angle information of the input graph, if its maximal degree $L$ is sufficiently large. For concision and without losing the generality, we assume the steerable features $\tilde{\boldsymbol{v}}_i^{(1)}$ are initialized with only spherical harmonics without the weights $\varphi_{\tilde{\boldsymbol{v}}, \text{init}}^{(l)}$ in Eq. (5). Let $\vec{\boldsymbol{x}}_{is} = (\vec{\boldsymbol{x}}_i - \vec{\boldsymbol{x}}_s)/\|\vec{\boldsymbol{x}}_i - \vec{\boldsymbol{x}}_s\|$, the inner product $z_{ij}^{(l)}$ can be expanded as follows

$$\left\langle \sum_{s \in \mathcal{N}(i)} Y^{(l)}(\vec{\boldsymbol{x}}_{is}), \sum_{t \in \mathcal{N}(j)} Y^{(l)}(\vec{\boldsymbol{x}}_{jt}) \right\rangle = \frac{4\pi}{2l+1} \sum_{s \in \mathcal{N}(i)} \sum_{t \in \mathcal{N}(j)} P^{(l)}(\langle \vec{\boldsymbol{x}}_{is}, \vec{\boldsymbol{x}}_{jt} \rangle), \tag{10}$$

where $P^{(l)} : \mathbb{R} \to \mathbb{R}$ is Legendre polynomial of degree $l$, and Eq. (10) is based on the properties of spherical harmonics that $\langle Y^{(l)}(\vec{\boldsymbol{x}}), Y^{(l)}(\vec{\boldsymbol{y}}) \rangle = 4\pi/(2l+1) \cdot P^{(l)}(\langle \vec{\boldsymbol{x}}, \vec{\boldsymbol{y}} \rangle), \|\vec{\boldsymbol{x}}\| = \|\vec{\boldsymbol{y}}\| = 1$. We have the following result.

**Theorem 4.1.** *For any geometric graph, there exists a bijection between the set of inner products $\{z_{ij}^{(l)}\}_{l=1}^{|\mathbb{A}_{ij}|}$ given by Eq. (10) and the set of edge angles $\mathbb{A}_{ij} = \{\theta_{is,jt} := \arccos\langle \vec{\boldsymbol{x}}_{is}, \vec{\boldsymbol{x}}_{jt} \rangle\}_{s \in \mathcal{N}(i), t \in \mathcal{N}(j)}$.*

Theorem 4.1 states that the inner products of full degrees can recover the information of all angles between each pair of edges, affirming the enhanced expressivity of our HEGNN. The proof is derived mainly based on the fact that Legendre polynomials are orthogonal polynomial bases which can injectivly represent the set $\mathbb{A}_{ij}$ thanks to Newton's identities. The details are deferred to the appendix. Although the upper-bound of the degree in Theorem 4.1 grows rapidly with the graph size, it will be shown in our experiments that HEGNN with only $L \leq 6$ is sufficient to outperform traditional models like EGNN [1] and TFN [19] in practice.

## 5 Experiment

### 5.1 Expressivity on Symmetric Graphs

**Design of experiments:** To experimentally verify the conclusion we proved above, we design a more comprehensive experiment based on code[5] in [35]. This experiment uses four $k$-fold structures ($k \in \{2, 3, 5, 10\}$) and five convex regular polyhedra shown in Fig. 1 as test objects, and the center of each is at the origin. In detail, an arbitrary rotation in $3D$ is acted on such symmetric structures called $\mathcal{G}_0$ which ensures the geometric graph after rotation called $\mathcal{G}_1$ does not coincide with the original one. The goal of our experiments is to check whether different equivariant neural networks can distinguish $\mathcal{G}_0$ and $\mathcal{G}_1$.

The models we select include two models that only use Cartesian coordinates: EGNN and GVP-GNN; and two models that use high-degree steerable features: TFN and MACE. However, TFN and MACE (denoted as TFN/MACE$_{l \leq L}$) always use all degrees $l \in \{0, \ldots, L\}$, so it is not clear which

---

[5]https://github.com/chaitjo/geometric-gnn-dojo.

degree(s) of steerable features distinguish the two geometric graphs. In our HEGNN, all steerable features corresponding to unwanted degrees could be masked during initialization in Eq. (5), and we let HEGNN$_{l=L}$ be a HEGNN with only $l$th-degree steerable features. Additionally, to align with TFN/MACE, we also test the performance of HEGNN with all $l \in \{0, 1, \ldots, L\}$ donated as HEGNN$_{l \leq L}$. Following the settings[6] in [35], the output of each graph is the concatenation of invariant scalars, coordinates, and high-degree steerable features pooling among all nodes. We then map this spliced vector to a two-dimensional vector and input it into a simple classifier to determine whether the equivariant graph neural network can distinguish $\mathcal{G}_0$ from $\mathcal{G}_1$.

**Results:** The results on $k$-fold are deferred to Appendix for saving space, and the results on regular polyhedra are shown in Table 2. From Table 1, we can know steerable features in which degree could not distinguish specific symmetry structure, and both results on $k$-fold and regular polyhedra are also in perfect agreement with our conclusions. Models (EGNN and GVP-GNN) only with Cartesian vectors cannot distinguish any symmetric graph at all. Taking HEGNN$_{l=5}$ as an example, since $\boldsymbol{D}^{(5)}(\mathfrak{H}) = 0, \forall \mathfrak{H} \in \{T, O, I\}$, no matter which kind of regular polyhedron, $f^{(5)}$ could only output 0 thus failing to distinguish the structures.

Table 2: *Regular polyhedra.*

| | GNN Layer | Tetrahedron | Cube | Octahedron | Dodecahedron | Icosahedron |
| --- | --- | --- | --- | --- | --- | --- |
| | | | | **Rotational symmetry** | | |
| Cart. | E-GNN$_{l=1}$ | 50.0 ± 0.0 | 50.0 ± 0.0 | 50.0 ± 0.0 | 50.0 ± 0.0 | 50.0 ± 0.0 |
| | GVP-GNN$_{l=1}$ | 50.0 ± 0.0 | 50.0 ± 0.0 | 50.0 ± 0.0 | 50.0 ± 0.0 | 50.0 ± 0.0 |
| Single Type Spherical | HEGNN$_{l=1}$ | 50.0 ± 0.0 | 50.0 ± 0.0 | 50.0 ± 0.0 | 50.0 ± 0.0 | 50.0 ± 0.0 |
| | HEGNN$_{l=2}$ | 50.0 ± 0.0 | 50.0 ± 0.0 | 50.0 ± 0.0 | 50.0 ± 0.0 | 50.0 ± 0.0 |
| | HEGNN$_{l=3}$ | **100.0** ± 0.0 | 50.0 ± 0.0 | 50.0 ± 0.0 | 50.0 ± 0.0 | 50.0 ± 0.0 |
| | HEGNN$_{l=4}$ | **100.0** ± 0.0 | 90.0 ±30.0 | 90.0 ±30.0 | 50.0 ± 0.0 | 50.0 ± 0.0 |
| | HEGNN$_{l=5}$ | 50.0 ± 0.0 | 50.0 ± 0.0 | 50.0 ± 0.0 | 50.0 ± 0.0 | 50.0 ± 0.0 |
| | HEGNN$_{l=6}$ | **100.0** ± 0.0 | **100.0** ± 0.0 | **100.0** ± 0.0 | **100.0** ± 0.0 | **100.0** ± 0.0 |
| | HEGNN$_{l=7}$ | **100.0** ± 0.0 | 50.0 ± 0.0 | 50.0 ± 0.0 | 50.0 ± 0.0 | 50.0 ± 0.0 |
| | HEGNN$_{l=8}$ | **100.0** ± 0.0 | 90.0 ±30.0 | 90.0 ±30.0 | 50.0 ± 0.0 | 50.0 ± 0.0 |
| | HEGNN$_{l=9}$ | **100.0** ± 0.0 | 50.0 ± 0.0 | 50.0 ± 0.0 | 50.0 ± 0.0 | 50.0 ± 0.0 |
| | HEGNN$_{l=10}$ | **100.0** ± 0.0 | **100.0** ± 0.0 | 95.0 ±15.0 | **100.0** ± 0.0 | **100.0** ± 0.0 |
| | HEGNN$_{l=11}$ | **100.0** ± 0.0 | 50.0 ± 0.0 | 50.0 ± 0.0 | 50.0 ± 0.0 | 50.0 ± 0.0 |
| Sph. | HEGNN/TFN/MACE$_{l \leq 2}$ | 50.0 ± 0.0 | 50.0 ± 0.0 | 50.0 ± 0.0 | 50.0 ± 0.0 | 50.0 ± 0.0 |
| | HEGNN/TFN/MACE$_{l \leq 3}$ | **100.0** ± 0.0 | 50.0 ± 0.0 | 50.0 ± 0.0 | 50.0 ± 0.0 | 50.0 ± 0.0 |
| | HEGNN/TFN/MACE$_{l \leq 4}$ | **100.0** ± 0.0 | **100.0** ± 0.0 | **100.0** ± 0.0 | 50.0 ± 0.0 | 50.0 ± 0.0 |
| | HEGNN/TFN/MACE$_{l \leq 6}$ | **100.0** ± 0.0 | **100.0** ± 0.0 | **100.0** ± 0.0 | **100.0** ± 0.0 | **100.0** ± 0.0 |

## 5.2 Physical Dynamics Simulation

**Datasets:** We benchmark our HEGNN in two scenarios, including: $N$**-body system** [43] is a dataset generated from simulations. In our simulations, each system contains $N$ charged particles with random charge $c_i \in \{0, 1\}$, whose movements are driven by Coulomb forces. To verify the efficiency and effectiveness of our HEGNN on datasets of different sizes, we select $N$ from $\{5, 20, 50, 100\}$. We use 5000 samples for training, 2000 for validation, and 2000 for testing. The task is to estimate the positions of the $N$ particles after 1,000 timesteps. **MD17** [44] dataset contains trajectory data for eight molecules generated through molecular dynamics simulations. The goal of this experiment is to predict the future positions of the atoms based on their current state. We follow the dataset partitioning scheme from [45], splitting the dataset into 500/2000/2000 frame pairs for training, validation and testing, respectively. All experiments are run on a single NVIDIA A100-80G GPU.

**Baselines:** To demonstrate the advantages of our HEGNN over both models with scalarization techniques and models with high-degree steerable vectors at the same time, our baseline needs to consider the selection issues of both models simultaneously. Therefore, we select some representative models as baselines, including the invariant RF [46], the equivariant EGNN [1], TFN [19] and SE(3)-Tr. [20]. In addition, we select classical models such as Linear dynamics [1], the non-equivariant

---
[6]It should be noted here that because only $0 \sim 11$-th-degree spherical harmonics can be used in e3nn [39], we only measure the models with up to 11th-degree here, and in Appendix B.3, We have given a new verification method.

Message Passing Neural Network (MPNN) [47], the invariant SchNet [48], and the equivariant GVP-GNN [49] for the $N$-body experiments. For MD17 experiments, we also select GMN [45].

**Metrics: 1. Loss function:** We use Mean Squared Error (MSE) to measure the accuracy of the prediction results in both experiments. **2. Inference time:** Given that the $N$-body system we use contains data of varying sizes, we test the inference time of each model on this dataset. The inference time for each model is calculated relative to the benchmark, which is the inference time of EGNN at the corresponding scale.

**Results on $N$-Body systems:** The main results of $N$-body system simulation are presented in Table 3. From these results, we observe the following: **1. Overall performance**: Our HEGNN significantly outperforms other models across datasets of all sizes. Although EGNN [1] performs better than high-degree steerable models like TFN or $SE(3)$-Transformer in this task, our HEGNN is still better than EGNN, which show that the method of HEGNN introducing high-degree steerable features is more effective. **2. Stability**: Although the performance of the model (HEGNN$_{l<6}$) using high-degree steerable features declines slightly when the geometric graph is small, overall, HEGNN performs better than other models. **3. Inference time**: Our model's inference time is significantly faster than that of high-degree steerable models like TFN, reflecting the simplicity and efficiency of our HEGNN.

**Results on MD17:** The main results of MD17 experiment are shown in Table 4, with some data sourced from [45]. From these results, we draw the following insights: **1. Overall performance:** Our HEGNN outperforms other models on six out of eight molecules. The effect on the remaining two molecules is only not as good as GMN [45] and this is because GMN introduces additional knowledge such as chemical bonds. **2. Advantage of high-degree vectors:** Most of the best results are obtained on HEGNN$_{l\leq6}$, indicating that the use of high-degree steerable features can enhance model expression capabilities.

Table 3: MSE and time-consuming ratio with EGNN [1] on $N$-body system.

| | 5-body | | 20-body | | 50-body | | 100-body | |
| --- | --- | --- | --- | --- | --- | --- | --- | --- |
| | MSE $(\times10^{-2})$ | Relative Time | MSE $(\times10^{-2})$ | Relative Time | MSE $(\times10^{-2})$ | Relative Time | MSE $(\times10^{-2})$ | Relative Time |
| Linear | 7.72 | 0.01 | 10.12 | 0.02 | 11.81 | 0.02 | 12.69 | 0.01 |
| MPNN [47] | 1.80 | 0.49 | 2.50 | 0.51 | 2.96 | 0.50 | 3.55 | 0.45 |
| SchNet [48] | 11.31 | 2.93 | 17.72 | 6.24 | 22.14 | 31.63 | 22.14 | 27.04 |
| RF [46] | 1.51 | 0.54 | 3.41 | 0.65 | 4.75 | 0.67 | 5.72 | 0.49 |
| GVP-GNN [49] | 7.26 | 2.36 | 5.76 | 2.38 | 7.07 | 2.42 | 7.55 | 2.33 |
| EGNN [1] | 0.65 | 1.00 | 1.01 | 1.00 | 1.00 | 1.00 | 1.36 | 1.00 |
| TFN$_{l\leq2}$ | 1.49 | 2.69 | 1.86 | 3.19 | 2.20 | 2.87 | 3.42 | 6.58 |
| TFN$_{l\leq3}$ | 1.76 | 3.91 | 1.87 | 4.54 | 1.94 | 4.89 | OOM | - |
| SE(3)-Tr.$_{l\leq2}$ | 3.24 | 4.94 | 3.19 | 5.88 | 2.54 | 5.97 | 2.33 | 5.15 |
| HEGNN$_{l\leq1}$ | 0.52 | 1.77 | 0.79 | 1.84 | 0.88 | 1.60 | 1.13 | 1.45 |
| HEGNN$_{l\leq2}$ | **0.47** | 1.88 | **0.78** | 1.94 | 0.90 | 1.71 | 0.97 | 1.55 |
| HEGNN$_{l\leq3}$ | 0.48 | 2.11 | 0.80 | 2.23 | **0.84** | 1.84 | 0.94 | 1.61 |
| HEGNN$_{l\leq6}$ | 0.69 | 2.14 | 0.86 | 2.43 | 0.96 | 2.18 | **0.86** | 1.90 |

Table 4: Prediction error $(\times10^{-2})$ on MD17 dataset. Results averaged across 3 runs.

| | Aspirin | Benzene | Ethanol | Malonaldehyde | Naphthalene | Salicylic | Toluene | Uracil |
| --- | --- | --- | --- | --- | --- | --- | --- | --- |
| RF | $10.94_{\pm0.01}$ | $103.72_{\pm1.29}$ | $4.64_{\pm0.01}$ | $13.93_{\pm0.03}$ | $0.50_{\pm0.01}$ | $1.23_{\pm0.01}$ | $10.93_{\pm0.04}$ | $0.64_{\pm0.01}$ |
| EGNN | $14.41_{\pm0.15}$ | $62.40_{\pm0.53}$ | $4.64_{\pm0.01}$ | $13.64_{\pm0.01}$ | $0.47_{\pm0.02}$ | $1.02_{\pm0.02}$ | $11.78_{\pm0.07}$ | $0.64_{\pm0.01}$ |
| EGNNReg | $13.82_{\pm0.19}$ | $61.68_{\pm0.37}$ | $6.06_{\pm0.01}$ | $13.49_{\pm0.06}$ | $0.63_{\pm0.01}$ | $1.68_{\pm0.01}$ | $11.05_{\pm0.01}$ | $0.66_{\pm0.01}$ |
| GMN | $10.14_{\pm0.03}$ | **$48.12_{\pm0.40}$** | $4.83_{\pm0.01}$ | $13.11_{\pm0.03}$ | $0.40_{\pm0.01}$ | $0.91_{\pm0.01}$ | **$10.22_{\pm0.08}$** | $0.59_{\pm0.01}$ |
| TFN$_{l\leq2}$ | $12.37_{\pm0.18}$ | $58.48_{\pm1.98}$ | $4.81_{\pm0.04}$ | $13.62_{\pm0.08}$ | $0.49_{\pm0.01}$ | $1.03_{\pm0.02}$ | $10.89_{\pm0.01}$ | $0.84_{\pm0.02}$ |
| SE(3)-Tr.$_{l\leq2}$ | $11.12_{\pm0.06}$ | $68.11_{\pm0.67}$ | $4.74_{\pm0.13}$ | $13.89_{\pm0.02}$ | $0.52_{\pm0.01}$ | $1.13_{\pm0.02}$ | $10.88_{\pm0.06}$ | $0.79_{\pm0.02}$ |
| HEGNN$_{l\leq1}$ | $10.32_{\pm0.58}$ | $62.53_{\pm7.62}$ | $4.63_{\pm0.01}$ | **$12.85_{\pm0.01}$** | $0.38_{\pm0.01}$ | $0.90_{\pm0.05}$ | $10.56_{\pm0.10}$ | $0.56_{\pm0.02}$ |
| HEGNN$_{l\leq2}$ | $10.04_{\pm0.45}$ | $61.80_{\pm5.92}$ | $4.63_{\pm0.01}$ | **$12.85_{\pm0.01}$** | $0.39_{\pm0.01}$ | $0.91_{\pm0.06}$ | $10.56_{\pm0.05}$ | $0.55_{\pm0.01}$ |
| HEGNN$_{l\leq3}$ | $10.20_{\pm0.23}$ | $62.82_{\pm4.25}$ | $4.63_{\pm0.01}$ | **$12.85_{\pm0.02}$** | **$0.37_{\pm0.01}$** | $0.94_{\pm0.10}$ | $10.55_{\pm0.16}$ | **$0.52_{\pm0.01}$** |
| HEGNN$_{l\leq6}$ | **$9.94_{\pm0.07}$** | $59.93_{\pm5.21}$ | **$4.62_{\pm0.01}$** | **$12.85_{\pm0.01}$** | **$0.37_{\pm0.02}$** | **$0.88_{\pm0.02}$** | $10.56_{\pm0.33}$ | $0.54_{\pm0.01}$ |

## 5.3 Perturbation Experiment

In practical scenarios, slight perturbations (such as molecular vibrations) can disrupt strict symmetry, potentially mitigating the conclusions outlined in Theorems 3.5 and 3.6. We therefore designed this perturbation experiment for a simple study and were surprised to find that HEGNN can still bring better robustness through the introduction of high-degree steerable features.

**Design of experiments:** We take the tetrahedron as an example and compare the cases of EGNN, $HEGNN_{l=3}$, and $HEGNN_{l\leq 3}$ when adding noise perturbations with results in Table 5. Here, $\varepsilon$ represents the ratio of noise, and the modulus of the noise obeys $\mathcal{N}(0, \varepsilon \cdot \mathbb{E}[\|\vec{x} - \vec{x}_c\|] \cdot I)$.

Table 5: Results for perturbation experiment.

|  | $\varepsilon = 0.01$ | $\varepsilon = 0.05$ | $\varepsilon = 0.10$ | $\varepsilon = 0.50$ |
|---|---|---|---|---|
| EGNN | 50.0 ± 0.0 | 45.0 ± 15.0 | 65.0 ± 22.9 | 60.0 ± 20.0 |
| $HEGNN_{l=3}$ | **100.0** ± 0.0 | **100.0** ± 0.0 | **100.0** ± 0.0 | **100.0** ± 0.0 |
| $HEGNN_{l\leq 3}$ | **100.0** ± 0.0 | **100.0** ± 0.0 | **100.0** ± 0.0 | **100.0** ± 0.0 |

**Results:** It can be observed that the performance of EGNN is slightly improved in the presence of noise (from 50% when $\varepsilon = 0.01$ to 60% when $\varepsilon = 0.5$), while HEGNN demonstrates better robustness. Even though symmetry-breaking factors will make the geometric graph deviate from the symmetric state, the deviated graph is still roughly symmetric. In other words, the outputs of equivariant GNNs on the derived graphs keep close to zero if the degree value is chosen to be those in Table 1, which will still lead to defective performance.

## 6 Conclusion

In this paper, we challenged the prevailing notion that higher-degree steerable vectors are unnecessary for achieving expressivity in equivariant Graph Neural Networks (GNNs). Through rigorous theoretical analysis, we demonstrated that equivariant GNNs constrained to 1st-degree representations inevitably degenerate to zero functions when applied to symmetric structures, such as $k$-fold rotations and regular polyhedra. To address this limitation, we introduced HEGNN, a high-degree extension of the EGNN model. HEGNN enhances expressivity by integrating higher-degree steerable vectors while retaining the efficiency of the original model through a scalarization technique. Our extensive empirical evaluations on various datasets, including the symmetric toy dataset, $N$-body, and MD17, validate our theoretical predictions. HEGNN not only adheres to our theoretical insights but also exhibits significant performance improvements over existing models. These findings underscore the critical role of higher-degree representations in fully leveraging the potential of equivariant GNNs.

## 7 Acknowledgment

This work was jointly supported by the following projects: the National Science and Technology Major Project under Grant 2020AAA0107300, the National Natural Science Foundation of China (No. 62376276, No. 62172422); Beijing Nova Program (No. 20230484278); Beijing Outstanding Young Scientist Program (No. BJJWZYJH012019100020098), the Fundamental Research Funds for the Central Universities, and the Research Funds of Renmin University of China (23XNKJ19); Public Computing Cloud, Renmin University of China.

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

# A  Theoretical Details

## A.1  Equivariance/Invariance of HEGNN

In this section, we demonstrate the equivariance of our HEGNN. In order to further illustrate the connection between our HEGNN, EGNN, and TFN, a general proof is given here.

Table 6: Comparison between our HEGNN and the scalarization-based model representing EGNN [1], and the high-degree steerable model representing TFN [19]. HEGNN combines the scalarization trick of EGNN that only uses invariant scalars (0th degree steerable features) to interact between steerable features corresponding to different degrees, avoiding the high computational cost of using CG tensor products in TFN.

| | EGNN [1] | TFN [19] | HEGNN (Ours) |
|---|---|---|---|
| Msg | $\boldsymbol{m}_{ij} = \phi_{\boldsymbol{m}}(\boldsymbol{h}_i, \boldsymbol{h}_j, \boldsymbol{e}_{ij}, d_{ij}^2)$ $\vec{\boldsymbol{m}}_{ij} = \varphi_{\vec{\boldsymbol{x}}}(\boldsymbol{m}_{ij}) \cdot (\vec{\boldsymbol{x}}_i - \vec{\boldsymbol{x}}_j)$ | $\tilde{\boldsymbol{m}}_{ij}^{(\mathbb{L})} = \tilde{\boldsymbol{v}}_i^{(\mathbb{L})} \otimes_{\mathrm{cg}}^{\boldsymbol{W}(d_{ij})} Y^{(\mathbb{L})}\left(\frac{\vec{\boldsymbol{x}}_{ij}}{\|\vec{\boldsymbol{x}}_{ij}\|}\right)$ | $\boldsymbol{m}_{ij} = \varphi_{\boldsymbol{m}}(\boldsymbol{h}_i, \boldsymbol{h}_j, \boldsymbol{e}_{ij}, d_{ij}^2, \bigoplus_{l \in \mathbb{L}} z_{ij}^{(l)})$ $\vec{\boldsymbol{m}}_{ij} = \varphi_{\vec{\boldsymbol{x}}}(\boldsymbol{m}_{ij}) \cdot (\vec{\boldsymbol{x}}_i - \vec{\boldsymbol{x}}_j)$ $\tilde{\boldsymbol{v}}_{ij}^{(l)} = \varphi_{\tilde{\boldsymbol{v}}}^{(l)}(\boldsymbol{m}_{ij}) \cdot (\tilde{\boldsymbol{v}}_i^{(l)} - \tilde{\boldsymbol{v}}_j^{(l)})$ |
| Agg | $\boldsymbol{m}_i = \alpha_i \sum_{j \in \mathcal{N}(i)} \boldsymbol{m}_{ij}$ $\vec{\boldsymbol{m}}_i = \alpha_i \sum_{j \in \mathcal{N}(i)} \vec{\boldsymbol{m}}_{ij}$ | $\tilde{\boldsymbol{m}}_i^{(\mathbb{L})} = \alpha_i \sum_{j \in \mathcal{N}(i)} \tilde{\boldsymbol{m}}_{ij}^{(\mathbb{L})}$ | $\boldsymbol{m}_i = \alpha_i \sum_{j \in \mathcal{N}(i)} \boldsymbol{m}_{ij}$ $\vec{\boldsymbol{m}}_i = \alpha_i \sum_{j \in \mathcal{N}(i)} \vec{\boldsymbol{m}}_{ij}$ $\tilde{\boldsymbol{m}}_i^{(l)} = \alpha_i \sum_{j \in \mathcal{N}(i)} \tilde{\boldsymbol{m}}_{ij}^{(l)}$ |
| Upd | $\Delta \boldsymbol{h}_i = \varphi_{\boldsymbol{h}}(\boldsymbol{h}_i, \boldsymbol{m}_i)$ $\Delta \vec{\boldsymbol{x}}_i = \vec{\boldsymbol{m}}_i$ | $\Delta \boldsymbol{v}_i^{(\mathbb{L})} = \boldsymbol{m}_i^{(\mathbb{L})}$ | $\Delta \boldsymbol{h}_i = \varphi_{\boldsymbol{h}}(\boldsymbol{h}_i, \boldsymbol{m}_i)$ $\Delta \vec{\boldsymbol{x}}_i = \vec{\boldsymbol{m}}_i$ $\Delta \tilde{\boldsymbol{v}}_i^{(l)} = \tilde{\boldsymbol{m}}_i^{(l)}$ |

**Theorem A.1** (Equivariance/Invariance of HEGNN). *$\boldsymbol{h}_i, \tilde{\boldsymbol{v}}_i^{(0)}$ in HEGNN is $\mathrm{E}(3)$ invariant, $\vec{\boldsymbol{x}}_i$ is $\mathrm{E}(3)$ equivariant. In addition, all $\tilde{\boldsymbol{v}}_i^{(l)}$ are $O(3)$ equivariant and translation invariant when $l$ is odd; all $\tilde{\boldsymbol{v}}_i^{(l)}$ is $SO(3)$ equivariant and inversion/translation invariant to when $l$ even.*

*Proof.* Consider a sequence composed of functions $\{\varphi_i : \mathcal{X}^{(i-1)} \to \mathcal{X}^{(i)}\}_{i=1}^N$ equivariant to a same group $\mathfrak{H}$, the equivariance lead to an interesting property that

$$\varphi_N \circ \cdots \circ \varphi_{i+1} \circ \rho_{\mathcal{X}^{(i)}}(\mathfrak{h}) \varphi_i \circ \cdots \circ \varphi_1 = \varphi_N \circ \cdots \circ \varphi_{j+1} \circ \rho_{\mathcal{X}^{(j)}}(\mathfrak{h}) \varphi_j \circ \cdots \circ \varphi_1,$$

holds for all $i, j \in \{1, 2, \ldots, N\}$ and $\mathfrak{h} \in \mathfrak{H}$, which means that the group elements $\mathfrak{h}$ can be freely exchanged in the composite sequence of equivariant functions. In particular, if one of the equivariant functions (*e.g.* $\varphi_k$) is replaced by an invariant function, the group element $\mathfrak{h}$ will be absorbed, which means

$$\varphi_N \circ \cdots \circ \varphi_k \circ \cdots \circ \varphi_{i+1} \circ \rho_{\mathcal{X}^{(i)}}(\mathfrak{h}) \varphi_i \circ \cdots \circ \varphi_1 = \varphi_N \circ \cdots \circ \varphi_1.$$

holds for all $\mathfrak{h} \in \mathfrak{H}$ but only $i \in \{1, 2, \ldots, k\}$. Although $\varphi_N \circ \cdots \circ \varphi_k$ is still equivariant, because the group elements must be input starting from $\varphi_1$, the overall $\varphi_N \circ \cdots \circ \varphi_1$ is still an invariant function. That is to say, to conclude that the entire HEGNN is equivariant, we only need to prove that HEGNN is equivariant in initialization and each layer.

The initialization of HEGNN is based on spherical harmonics, which is similar to TFN. Spherical harmonics are inherently equivariant, that is,

$$Y^{(l)}(\boldsymbol{R}_{\mathfrak{r}} \vec{\boldsymbol{x}}) = \boldsymbol{D}^{(l)}(\mathfrak{r}) Y^{(l)}(\vec{\boldsymbol{x}}).$$

Note that variables participating in the coefficient in Eq. (5) are all invariant scalars, so the initialization of HEGNN is consistent with the spherical harmonic function. Note that for Cartesian vectors, they can be aligned by arranging the spherical harmonics of 1st degree [19], that is,

$$\vec{\boldsymbol{x}} \propto Y^{(1)}(\vec{\boldsymbol{x}}).$$

From this perspective, EGNN can also be considered to be initialized using spherical harmonics, but this step is omitted because the value is proportional to the input Cartesian vector.

It is worth explaining that spherical harmonics are inversion invariant when $l$ is even, that is,

$$Y^{(l)}(\mathfrak{m}\vec{\boldsymbol{x}}) = Y^{(l)}(\vec{\boldsymbol{x}}).$$

Equivariant ones are not necessarily better than invariant ones. When we need to predict pseudovectors (such as moments), we need to inversion invariant 1st-degree steerable features, because pseudovectors are inversion invariant. This is why introducing the cross product $\vec{x} \times \vec{y}$ (the result is inversion invariant) into a linear combination can only build a $\mathrm{SE}(3)$ equivariant network, but not a $\mathrm{E}(3)$ network [50].

In fact, inversion equivariant/invariant high degree steerable features can be obtained by calculating the CG tensor product of spherical harmonics and inversion equivariant Cartesian vectors like $\vec{x}_i - \vec{x}_c$ [39]. Moreover, the equivariance at each layer is also easy to prove and the internal Wigner-D matrix can be extracted through the CG tensor product [31].

$$D^{(l)}(\mathfrak{h})\tilde{v}^{(l)} = \left[\left(D^{(l_1)}(\mathfrak{h})\tilde{v}^{(l_1)}\right) \otimes_{\mathrm{cg}}^{W} \left(D^{(l_2)}(\mathfrak{h})\tilde{v}^{(l_2)}\right)\right]^{(l)}.$$

When the output result is an invariant scalar, the Wigner-D matrix degenerates into a trivial representation 1. Norm and inner product are all special cases of this type, so equivariance is established. From this perspective, EGNN and HEGNN are equivalent to using only the weight coefficients of $(l, l) \to 0$ and $(0, l) \to l$. Similar ideas include the steerable MLPs in [33] . $\square$

## A.2 Other Proofs

**Theorem A.2** (Theorem 3.4). *Suppose that $f^{(l)}$ is an $\mathrm{O}(3)$-equivariant function on geometric graphs, regarding the group representation $\rho^{(l)}$ defined in Eq. (2). Then, for any symmetric graph $\mathcal{G}$ induced by the group $\mathfrak{H} \leq \mathrm{O}(3)$, namely, $\forall \mathcal{G} \in \mathbb{G}(\mathfrak{H})$, we always have*

$$f^{(l)}(\mathcal{G}) = \rho^{(l)}(\mathfrak{H})f^{(l)}(\mathcal{G}). \tag{11}$$

*Here we have defined group average as $\rho^{(l)}(\mathfrak{H}) := \frac{1}{|\mathfrak{H}|}\sum_{\mathfrak{h}\in\mathfrak{H}}\rho^{(l)}(\mathfrak{h})$.*

*Proof.* If $f^{(l)}$ is $\mathrm{O}(3)$-equivariant, then it is also a $\mathfrak{H}$-equivariant, thus

$$f^{(l)}(\mathcal{G}) = \frac{1}{|\mathfrak{H}|}\sum_{\mathfrak{h}\in\mathfrak{H}}f^{(l)}(\mathfrak{h}\cdot\mathcal{G}) = \frac{1}{|\mathfrak{H}|}\sum_{\mathfrak{h}\in\mathfrak{H}}\rho(\mathfrak{h})f^{(l)}(\mathcal{G}) = \left(\frac{1}{|\mathfrak{H}|}\sum_{\mathfrak{h}\in\mathfrak{H}}\rho(\mathfrak{h})\right)f^{(l)}(\mathcal{G}) = \rho(\mathfrak{H})f^{(l)}(\mathcal{G}).$$

$\square$

**Theorem A.3** (Theorem 3.5). *If and only if the matrix $I_{2l+1} - \rho^{(l)}(\mathfrak{H})$ is non-singular, the $\mathrm{O}(3)$-equivariant function $f^{(l)}$ is always a zero function on $\mathcal{G}$, namely,*

$$f^{(l)}(\mathcal{G}) \equiv \mathbf{0}, \quad \forall \mathcal{G} \in \mathbb{G}(\mathfrak{H}). \tag{12}$$

*Proof.* From Theorem 3.4, we know that

$$f^{(l)}(\mathcal{G}) = \rho(\mathfrak{H})f^{(l)}(\mathcal{G}) \iff \left(I_{2l+1} - \rho^{(l)}(\mathfrak{H})\right)f^{(l)}(\mathcal{G}) = 0,$$

and the theorem holds for basic knowledge of linear algebra. $\square$

**Theorem A.4** (Theorem 3.6). *For a finite group $\mathfrak{H}$ with its representation $\rho^{(l)}$, $\rho^{(l)}(\mathfrak{H})$ is a zero matrix (i.e., $\rho^{(l)}(\mathfrak{H}) = \mathbf{0}$) if and only if $\mathrm{tr}(\rho^{(l)}(\mathfrak{H})) = 0$. In this case, $f^{(l)}(\mathcal{G}) \equiv \mathbf{0}, \forall \mathcal{G} \in \mathbb{G}(\mathfrak{H})$.*

*Proof.* It is obvious that $\rho^{(l)}(\mathfrak{H}) = \mathbf{0} \implies \mathrm{tr}(\rho^{(l)}(\mathfrak{H})) = 0$ since all elements are zero not to mention the main diagonal, and we only to prove $\mathrm{tr}(\rho^{(l)}(\mathfrak{H})) = 0 \implies \rho^{(l)}(\mathfrak{H}) = \mathbf{0}$.

A basic fact is $\mathfrak{h} \cdot \mathfrak{H} = \mathfrak{H}$, thereby

$$\rho^{(l)}(\mathfrak{h})\rho^{(l)}(\mathfrak{H}) = \rho^{(l)}(\mathfrak{H}).$$

Now we can use group average and get

$$\left(\rho^{(l)}(\mathfrak{H})\right)^2 = \rho^{(l)}(\mathfrak{H})$$

Such operation can be repeated many times, so that

$$\left(\rho^{(l)}(\mathfrak{H})\right)^k = \rho^{(l)}(\mathfrak{H}), \qquad \forall k \in \mathbb{N}_+.$$

Now we calculate the trace for each matrix and find

$$\mathrm{tr}\left(\left(\rho^{(l)}(\mathfrak{H})\right)^k\right) = \mathrm{tr}\left(\rho^{(l)}(\mathfrak{H})\right) = 0, \qquad \forall k \in \mathbb{N}_+.$$

By Newton's identity [51], all eigenvalues of the matrix $\rho(\mathfrak{H})$ are 0, that is, the matrix is a zero matrix. $\qquad\square$

Table 7: The traces of symmetric groups based on [52]. Trace for polyhedral groups can be calculated by $\boldsymbol{D}^{(l)}(H) = \lfloor l/r \rfloor + b[l \bmod r]$ with repeat length $r$, where $b$ is a string only with 0 or 1. For exmaple, for Tetrahedral group $T$, $\boldsymbol{D}^{(5)}(T) = \lfloor 5/6 \rfloor + b[5 \bmod 6] = 0 + 0 = 0$.

| Group | Notation | Data for Wigner-D matrix traces $\boldsymbol{D}^{(l)}(H)$ | |
|---|---|---|---|
| Reflection group | $C_i$ | $(2l+1) \cdot \delta_{l \bmod 2,0}$ | |
| Cyclic group | $C_n$ | $2\lfloor l/n \rfloor + 1$ | |
| Dihedral group | $D_n$ | $\lfloor l/n \rfloor + \delta_{l \bmod 2,0}$ | |
| Tetrahedral group | $T$ | $r = 6$ | $b = 100110$ |
| Octahedral group | $O$ | $r = 12$ | $b = 100010101110$ |
| Icosahedral group | $I$ | $r = 30$ | $b = 100000100010100110101110111110$ |

**Theorem A.5** (Theorem 4.1). *For any geometric graph, there exists a bijection between the set of inner products $\{z_{ij}^{(l)}\}_{l=1}^{|\mathbb{A}_{ij}|}$ given by Eq. (10) and the set of edge angles $\mathbb{A}_{ij} = \{\theta_{is,jt} := \langle \vec{x}_{is}, \vec{x}_{jt} \rangle\}_{s \in \mathcal{N}(i), t \in \mathcal{N}(j)}$.*

*Proof.* Note that the Legendre polynomial is a set of orthogonal polynomial bases, and there is a bijection to the power function polynomial space, that is

$$\mathrm{span}\left\{\sum_{n=1}^{|\mathbb{A}_{ij}|} P^{(l)}(\cos\theta_n)\right\}_{l=0}^M = \mathrm{span}\left\{\sum_{n=1}^{|\mathbb{A}_{ij}|} \cos^\alpha \theta_n\right\}_{\alpha=0}^M,$$

where $M$ is any non-negative integer represents the degree of the polynomial space. Moreover, from the knowledge of Newton's identities, the space of power sums can be converted to space of elementary symmetric polynomials as

$$\mathrm{span}\left\{\sum_{n=1}^{|\mathbb{A}_{ij}|} \cos^\alpha \theta_n\right\}_{\alpha=0}^M = \mathrm{span}\left\{\sum_{1 \leq n_1 < n_2 < \ldots n_m \leq |A_{ij}|} \left(\prod_{\nu=1}^k \cos\theta_{n_k}\right)\right\}_{\alpha=0}^M$$

From Vieta's formulas, when $M = |\mathbb{A}_{ij}|$, with the $M+1$ polynomial in the space of elementary symmetric polynomials being coefficients, we can build a $|\mathbb{A}_{ij}|$-degree polynomial with $\{\cos\theta \mid \theta \in \mathbb{A}_{ij}\}$ as its all roots[7]. Since all angles are in $[0, \pi)$, the cosine uniquely determines the angle value, and the proposition is established. $\qquad\square$

### A.3 Further Discussion

Our theory in fact shows that the degeneration of global features of a certain degree (in Table 1) are *inevitable* on symmetric geometric graphs. This raises two points worth discussing:

1. The degree not indicated to degenerate not necessarily produce a non-zero representation, which may still be affected by the model form and the edge situation.
2. There are some tricks to get around this degeneration: for example, making the output a set or relaxing equivariance constraints (*e.g.* probabilistic symmetry breaking [54]).

---

[7]The trick comes from Lemma 6 in DeepSet [53].

It is worth mentioning that outputting a set can solve most problems, although such operators may be quite intractable to implement in computer systems. The failure cases of frame averaging [26; 27] and Neural P$^3$M [30] which depend on singular value decomposition or eigenvalue decomposition, is caused by the non-unique matrix decomposition. Some other examples include ComENet [55], which uses the `scatter_min()` operator in PyTorch to extract the nearest neighbors, making it intractable to handle the situation where multiple neighbors are simultaneously closest, which is quite common in chemical molecules (*e.g.* -CH$_3$, -NH$_2$).

Moreover, from Theorem 4.1, we show the expressivity of our HEGNN, which is able to recover the information of all angles between each pair of edge. However, it should be noted that $|\mathbb{A}_{ij}|$ may be an extremely large number, which is unacceptable in practical applications to achieve completeness. The same problem also arises in discussions based on CG tensor product models (*e.g.* TFN [19]), such as discussions based on $D$-spanning [23], because a sufficiently high-degree $D$ is unacceptable. However, in terms of actual results, the performance of both our HEGNN and TFN is remarkable. From this perspective, how to bridge the gap between completeness and actual performance with features of limited channels and degrees is a question worth considering.

To get the ball rolling, we raise an interesting question here. Is there a performance gap between this type of purely mathematical representation and other features based on physical and biochemical prior knowledge?

- Purely mathematical representation: topological characteristics [56–58], cluster expansion basis [59; 60], subgraph blocks [61–63], frames [26; 27; 64], normalization operators [65; 66];

- Features based on physical and biochemical prior knowledge: distance matrix [67–72], chemical bond length, angle and dihedral angle [55; 73–75], force [76; 77], fractional coordinates [78–80], canonical ordering [81; 82].

In fact, some of these features are directly related (*e.g.* frames and fractional coordinates). How to construct effective and interpretable pure mathematical features based on those with prior knowledge will become a key point in network design, and we will consider further exploration in future work.

## B  More Experimental Details and Results

### B.1  Comparison of parameters between models

Like EGNN [1], different features use different numbers of channels, so the inference time does not obviously reflect the time complexity of $\mathcal{O}(L^2)$. We list the details of our HEGNN of different degree in Table 8. Intuitively, we add one of each steerable feature on the basis of EGNN (using 64 invariant scalars and 2 Cartesian vectors, *i.e.* coordinate and velocity).

Table 8: Channels for steerable features of different degrees and total dimensions of HEGNN of different degrees.

| | HEGNN$_{l\leq 1}$ | HEGNN$_{l\leq 2}$ | HEGNN$_{l\leq 3}$ | HEGNN$_{l\leq 6}$ |
|---|---|---|---|---|
| Channel for $\tilde{\boldsymbol{v}}^{(0)}$ | 65 | 65 | 65 | 65 |
| Channel for $\tilde{\boldsymbol{v}}^{(1)}$ | 3 | 3 | 3 | 3 |
| Channel for $\tilde{\boldsymbol{v}}^{(2)}$ | – | 1 | 1 | 1 |
| Channel for $\tilde{\boldsymbol{v}}^{(3)}$ | – | – | 1 | 1 |
| Channel for $\tilde{\boldsymbol{v}}^{(4)}$ | – | – | – | 1 |
| Channel for $\tilde{\boldsymbol{v}}^{(5)}$ | – | – | – | 1 |
| Channel for $\tilde{\boldsymbol{v}}^{(6)}$ | – | – | – | 1 |
| Total dimensions | 74 | 79 | 86 | 119 |

Table 9 shows the number of parameters and inference time of different models.

Table 9: Parameters and inference time (on 100-body dataset) of EGNN and other high-degree models.

| | Parameters | | | |
|---|---|---|---|---|
| | $l \leq 1$ | $l \leq 2$ | $l \leq 3$ | $l \leq 6$ |
| EGNN | 134.1k | – | – | – |
| HEGNN | 160.3k | 160.9k | 161.5k | 163.2k |
| TFN | 3.7M | 9.6M | 19.5M | 86.6M |
| SEGNN | 228.1k | 244.1k | 254.9k | 288.1k |
| MACE | 14.8M | 38.0M | 77.0M | 342.5M |
| | Inference Time ($10^{-2}$s) | | | |
| | $l \leq 1$ | $l \leq 2$ | $l \leq 3$ | $l \leq 6$ |
| EGNN | 0.57 | – | – | – |
| HEGNN | 0.82 | 0.88 | 0.91 | 1.08 |
| TFN | – | 3.75 | 2.66 | OOM |
| SEGNN | 1.33 | 1.7 | 1.98 | 22.33 |
| MACE | 22.10 | 125.87 | 261.11 | OOM |

## B.2 Experiment on $k$-fold Structure.

The results on the $k$-fold structure are completely consistent with the conclusion of Table 1. The few results that cannot reach 100% are due to the small number of training epochs, so the classifier fails to perform perfect classification. In fact, the accuracy of models can achieve $100.00_{\pm 0.0}$ after increasing the number of training rounds of the model like 500 epochs. Since from 2nd-degree steerable features can distinguish $\mathcal{G}_0$ and $\mathcal{G}_1$, and HEGNN/TFN/MACE$_{l \leq L}$ contain 2nd-degree steerable features when $l \geq 3$, the extra results are hidden but all their values are $100.00_{\pm 0.0}$.

Table 10: *$k$-fold symmetric structures.*

| | GNN Layer | Rotational symmetry | | | |
|---|---|---|---|---|---|
| | | 2 fold | 3 fold | 5 fold | 10 fold |
| Cart. | E-GNN$_{l=1}$ | $50.0_{\pm 0.0}$ | $50.0_{\pm 0.0}$ | $50.0_{\pm 0.0}$ | $50.0_{\pm 0.0}$ |
| | GVP-GNN$_{l=1}$ | $50.0_{\pm 0.0}$ | $50.0_{\pm 0.0}$ | $50.0_{\pm 0.0}$ | $50.0_{\pm 0.0}$ |
| Single Type Spherical | HEGNN$_{l=1}$ | $50.0_{\pm 0.0}$ | $50.0_{\pm 0.0}$ | $50.0_{\pm 0.0}$ | $50.0_{\pm 0.0}$ |
| | HEGNN$_{l=2}$ | $95.0_{\pm 15.0}$ | $95.0_{\pm 15.0}$ | $95.0_{\pm 15.0}$ | $95.0_{\pm 15.0}$ |
| | HEGNN$_{l=3}$ | $50.0_{\pm 0.0}$ | $75.0_{\pm 40.31}$ | $50.0_{\pm 0.0}$ | $50.0_{\pm 0.0}$ |
| | HEGNN$_{l=4}$ | $\mathbf{100.0_{\pm 0.0}}$ | $95.0_{\pm 15.0}$ | $95.0_{\pm 15.0}$ | $95.0_{\pm 15.0}$ |
| | HEGNN$_{l=5}$ | $50.0_{\pm 0.0}$ | $\mathbf{100.0_{\pm 0.0}}$ | $\mathbf{100.0_{\pm 0.0}}$ | $50.0_{\pm 0.0}$ |
| | HEGNN$_{l=6}$ | $\mathbf{100.0_{\pm 0.0}}$ | $95.0_{\pm 15.0}$ | $\mathbf{100.0_{\pm 0.0}}$ | $95.0_{\pm 15.0}$ |
| | HEGNN$_{l=7}$ | $50.0_{\pm 0.0}$ | $95.0_{\pm 15.0}$ | $90.0_{\pm 30.0}$ | $50.0_{\pm 0.0}$ |
| | HEGNN$_{l=8}$ | $\mathbf{100.0_{\pm 0.0}}$ | $90.0_{\pm 30.0}$ | $85.0_{\pm 32.02}$ | $85.0_{\pm 32.02}$ |
| | HEGNN$_{l=9}$ | $50.0_{\pm 0.0}$ | $\mathbf{100.0_{\pm 0.0}}$ | $\mathbf{100.0_{\pm 0.0}}$ | $50.0_{\pm 0.0}$ |
| | HEGNN$_{l=10}$ | $\mathbf{100.0_{\pm 0.0}}$ | $\mathbf{100.0_{\pm 0.0}}$ | $\mathbf{100.0_{\pm 0.0}}$ | $\mathbf{100.0_{\pm 0.0}}$ |
| | HEGNN$_{l=11}$ | $50.0_{\pm 0.0}$ | $\mathbf{100.0_{\pm 0.0}}$ | $95.0_{\pm 15.0}$ | $50.0_{\pm 0.0}$ |
| Sph. | HEGNN/TFN/MACE$_{l \leq 1}$ | $50.0_{\pm 0.0}$ | $50.0_{\pm 0.0}$ | $50.0_{\pm 0.0}$ | $50.0_{\pm 0.0}$ |
| | HEGNN/TFN/MACE$_{l \leq 2}$ | $\mathbf{100.0_{\pm 0.0}}$ | $\mathbf{100.0_{\pm 0.0}}$ | $\mathbf{100.0_{\pm 0.0}}$ | $\mathbf{100.0_{\pm 0.0}}$ |

## B.3 Further Verification on Regular Polyhedra

**Implementation details:** Since the e3nn [39] library only implements spherical harmonics up to the 11th-degree, we verified the expressivity of our HEGNN in disguise. In this experiment, we measure the expressivity of our HEGNN by calculating whether the high-degree feature $\tilde{\boldsymbol{v}}_i^{(l)}$ updates on regular polyhedra. Namely, if $\Delta \tilde{\boldsymbol{v}}_i^{(L)} = 0$, then HEGNN$_{l=L}$ loses the expressivity. This judgment is necessary and sufficient. We calculated the sum of all degree vectors from $L = 1$ to 30, which

covers the maximum repeat length $r$ in Table 7. We implemented the calculation of the high-degree features based on `scipy` [83] library.

**Results:** Our calculation results are shown in Table 11 which is completely consistent with the theoretical results in Table 7. The simple experiment shows that the sum of spherical harmonics $\sum_i Y^{(l)}(\vec{x}_i - \vec{x}_c)$, as a function on graph, will actually vanish on regular polyhedra for some integer degree $l$.

Table 11: *Expressivity analysis of HEGNN using sums of spherical harmonics.* Here, "True" indicates that our HEGNN can distinguish the orientations, meaning the norm of the sum of spherical harmonics is greater than 1. "False" in the table means that no distinction can be made, with the norm of the corresponding spherical harmonic being less than $10^{-3}$.

| | Rotational symmetry | | | | |
|---|---|---|---|---|---|
| $L$ | Tetrahedron | Cube | Octahedron | Dodecahedron | Icosahedron |
| 1 | False | False | False | False | False |
| 2 | False | False | False | False | False |
| 3 | True | False | False | False | False |
| 4 | True | True | True | False | False |
| 5 | False | False | False | False | False |
| 6 | True | True | True | True | True |
| 7 | True | False | False | False | False |
| 8 | True | True | True | False | False |
| 9 | True | False | False | False | False |
| 10 | True | True | True | True | True |
| 11 | True | False | False | False | False |
| 12 | True | True | True | True | True |
| 13 | True | False | False | False | False |
| 14 | True | True | True | False | False |
| 15 | True | False | False | False | False |
| 16 | True | True | True | True | True |
| 17 | True | False | False | False | False |
| 18 | True | True | True | True | True |
| 19 | True | False | False | False | False |
| 20 | True | True | True | True | True |
| 21 | True | False | False | False | False |
| 22 | True | True | True | True | True |
| 23 | True | False | False | False | False |
| 24 | True | True | True | True | True |
| 25 | True | False | False | False | False |
| 26 | True | True | True | True | True |
| 27 | True | False | False | False | False |
| 28 | True | True | True | True | True |
| 29 | True | False | False | False | False |
| 30 | True | True | True | True | True |

## B.4  Results on other dataset settings

The $N$-body dataset setting of this paper refers to FastEGNN [28][8], that is, 5,000 samples are used as the training set (instead of 3,000). We tested the situation of using different data segmentation in Table 12. We also add ClofNet [50], a local frame based scalarization method and SEGNN [33] (select $l = 1$ according to the optimal situation in the paper) and MACE [37] ($l = 2$), two classic high-degree steerable models for comparison.

---

[8] https://github.com/GLAD-RUC/FastEGNN.

Table 12: Results of $N$-body dataset under two partitions.

| $N$-body($\times 10^{-2}$) | 5-body | 20-body | 50-body | 100-body |
|---|---|---|---|---|
| train/valid/test=3k/2k/2k | | | | |
| EGNN | 0.71 | 1.08 | 1.16 | 1.29 |
| ClofNet | 0.89 | 1.79 | 2.40 | 2.94 |
| ClofNet-vel | 0.84 | 1.50 | 2.28 | 2.67 |
| GMN | 0.67 | 1.21 | 1.18 | 2.55 |
| MACE | 1.43 | 1.93 | 2.20 | 2.51 |
| SEGNN | 1.81 | 2.67 | 3.44 | NaN |
| HEGNN$_{l\leq 1}$ | 0.64 | **0.84** | 0.92 | 1.04 |
| HEGNN$_{l\leq 2}$ | 0.69 | 0.89 | 1.13 | **0.94** |
| HEGNN$_{l\leq 3}$ | **0.58** | 1.04 | **0.92** | 1.04 |
| HEGNN$_{l\leq 6}$ | 0.77 | 1.06 | 1.02 | 1.18 |
| train/valid/test=5k/2k/2k | | | | |
| ClofNet | 0.80 | 1.49 | 2.28 | 2.77 |
| ClofNet-vel | 0.78 | 1.45 | 2.22 | 2.77 |
| GMN | 0.52 | 0.98 | 1.04 | 1.21 |
| MACE | 1.13 | 1.60 | 2.41 | 3.38 |
| SEGNN | 1.68 | 2.63 | 3.30 | NaN |
| HEGNN$_{l\leq 1}$ | 0.52 | 0.79 | 0.88 | 1.13 |
| HEGNN$_{l\leq 2}$ | **0.47** | **0.78** | 0.90 | 0.97 |
| HEGNN$_{l\leq 3}$ | 0.48 | 0.80 | **0.84** | 0.94 |
| HEGNN$_{l\leq 6}$ | 0.69 | 0.86 | 0.96 | **0.86** |

Note that SEGNN in Table 12 does not show the effect in the original paper. We also tried to reproduce the dataset in the original paper [33], and the effect of SEGNN on 5-body dataset can be reproduced. However, see Table 13, it is also shown that SEGNN performs poorly on larger datasets.

Table 13: Comparison between EGNN, SEGNN and HEGNN on $N$-body from [33]

| $N$-body($\times 10^{-2}$) | 5-body | 20-body | 50-body | 100-body |
|---|---|---|---|---|
| EGNN | 0.71 | 1.04 | 1.15 | 1.31 |
| SEGNN | **0.50** | 6.61 | 9.34 | 13.46 |
| HEGNN$_{l\leq 1}$ | 0.71 | 0.97 | **0.93** | 1.22 |
| HEGNN$_{l\leq 2}$ | 0.65 | **0.91** | 1.05 | 1.14 |
| HEGNN$_{l\leq 3}$ | 0.63 | 0.99 | 1.05 | 1.27 |
| HEGNN$_{l\leq 6}$ | 0.72 | 1.05 | 1.11 | 1.28 |

We found that the GMN-L method proposed in [29] generally performs the best on MD17 dataset, but our HEGNN-6 also achieves comparable performance to GMN-L in most cases. Given that GMN-L requires careful handcrafting of constraints for chemical bonds into the model design, our model's ability to derive promising results without such enhancements supports its competitive performance.

Table 14: Prediction error ($\times 10^{-2}$) on MD17 dataset. Results averaged across 3 runs.

| | Aspirin | Benzene | Ethanol | Malonaldehyde | Naphthalene | Salicylic | Toluene | Uracil |
|---|---|---|---|---|---|---|---|---|
| RF | $10.94_{\pm 0.01}$ | $103.72_{\pm 1.29}$ | $4.64_{\pm 0.01}$ | $13.93_{\pm 0.03}$ | $0.50_{\pm 0.01}$ | $1.23_{\pm 0.01}$ | $10.93_{\pm 0.04}$ | $0.64_{\pm 0.01}$ |
| EGNN | $14.41_{\pm 0.15}$ | $62.40_{\pm 0.53}$ | $4.64_{\pm 0.01}$ | $13.64_{\pm 0.01}$ | $0.47_{\pm 0.02}$ | $1.02_{\pm 0.02}$ | $11.78_{\pm 0.07}$ | $0.64_{\pm 0.01}$ |
| EGNNReg | $13.82_{\pm 0.19}$ | $61.68_{\pm 0.37}$ | $6.06_{\pm 0.01}$ | $13.49_{\pm 0.06}$ | $0.63_{\pm 0.01}$ | $1.68_{\pm 0.01}$ | $11.05_{\pm 0.01}$ | $0.66_{\pm 0.01}$ |
| GMN | $10.14_{\pm 0.03}$ | $\mathbf{48.12}_{\pm 0.40}$ | $4.83_{\pm 0.01}$ | $13.11_{\pm 0.03}$ | $0.40_{\pm 0.01}$ | $0.91_{\pm 0.01}$ | $\mathbf{10.22}_{\pm 0.08}$ | $0.59_{\pm 0.01}$ |
| GMN-L | $\mathbf{9.76}_{\pm 0.11}$ | $\underline{54.17}_{\pm 0.69}$ | $4.63_{\pm 0.01}$ | $\mathbf{12.82}_{\pm 0.03}$ | $0.41_{\pm 0.01}$ | $\mathbf{0.88}_{\pm 0.01}$ | $\underline{10.45}_{\pm 0.04}$ | $0.59_{\pm 0.01}$ |
| TFN$_{l\leq 2}$ | $12.37_{\pm 0.18}$ | $58.48_{\pm 1.98}$ | $4.81_{\pm 0.04}$ | $13.62_{\pm 0.08}$ | $0.49_{\pm 0.01}$ | $1.03_{\pm 0.02}$ | $10.89_{\pm 0.01}$ | $0.84_{\pm 0.02}$ |
| SE(3)-Tr.$_{l\leq 2}$ | $11.12_{\pm 0.06}$ | $68.11_{\pm 0.67}$ | $4.74_{\pm 0.13}$ | $13.89_{\pm 0.02}$ | $0.52_{\pm 0.01}$ | $1.13_{\pm 0.02}$ | $10.88_{\pm 0.06}$ | $0.79_{\pm 0.02}$ |
| HEGNN$_{l\leq 1}$ | $10.32_{\pm 0.58}$ | $62.53_{\pm 7.62}$ | $\underline{4.63}_{\pm 0.01}$ | $\underline{12.85}_{\pm 0.01}$ | $\underline{0.38}_{\pm 0.01}$ | $\underline{0.90}_{\pm 0.05}$ | $10.56_{\pm 0.10}$ | $0.56_{\pm 0.02}$ |
| HEGNN$_{l\leq 2}$ | $10.04_{\pm 0.45}$ | $61.80_{\pm 5.92}$ | $\underline{4.63}_{\pm 0.01}$ | $\underline{12.85}_{\pm 0.01}$ | $0.39_{\pm 0.01}$ | $0.91_{\pm 0.06}$ | $10.56_{\pm 0.05}$ | $0.55_{\pm 0.01}$ |
| HEGNN$_{l\leq 3}$ | $10.20_{\pm 0.23}$ | $62.82_{\pm 4.25}$ | $\underline{4.63}_{\pm 0.01}$ | $\underline{12.85}_{\pm 0.02}$ | $\mathbf{0.37}_{\pm 0.01}$ | $0.94_{\pm 0.10}$ | $10.55_{\pm 0.16}$ | $\mathbf{0.52}_{\pm 0.01}$ |
| HEGNN$_{l\leq 6}$ | $\underline{9.94}_{\pm 0.07}$ | $59.93_{\pm 5.21}$ | $\mathbf{4.62}_{\pm 0.01}$ | $\underline{12.85}_{\pm 0.01}$ | $\mathbf{0.37}_{\pm 0.02}$ | $\mathbf{0.88}_{\pm 0.02}$ | $10.56_{\pm 0.33}$ | $\underline{0.54}_{\pm 0.01}$ |

## B.5 Expressiveness of initialization layer

Note that both EGNN [1] and HEGNN$_{l\leq 1}$ only use Cartesian vectors. However, in Table 3, the effect of the latter is greatly improved. We speculate that there are two possible factors for this improvement: 1) the extra layer of message passing brought in by Eq. (5); 2) the multi-channel of Cartesian vectors (see Table 8). Therefore, we tested the effect of HEGNN$_{l\leq 1}$-3layers, and the results are shown in Table 15. From the improvement, the first factor contributes more.

Table 15: Comparison between EGNN and HEGNN on $N$-body.

| $N$-body($\times 10^{-2}$) | 5-body | 20-body | 50-body | 100-body |
|---|---|---|---|---|
| EGNN-4layers | 0.65 | 1.01 | 1.00 | 1.36 |
| HEGNN$_{l\leq 1}$-3layers | 0.63 | 0.98 | 0.96 | 1.31 |
| HEGNN$_{l\leq 1}$-4layers | 0.52 | 0.79 | 0.88 | 1.13 |

## C  Limitation

Our current experiments are mainly limited to testing on small molecules and have not been verified on large-scale molecules or large-scale physical systems. Whether our HEGNN is effective on large-scale geometric graph data sets remains to be verified.

## D  Broader Impact

Our research belongs to the field of AI for Science. The HEGNN proposed in this article is expected to better model scientific problems, thereby promoting the development of higher-precision and efficient AI scientific models.

