# OpenReview forum: "Are High-Degree Representations Really Unnecessary in Equivariant Graph Neural Networks?"
_NeurIPS.cc/2024/Conference — NeurIPS 2024 poster_

### Official Review · Reviewer_3KcU · 2024-07-11

**Soundness:** 4
**Presentation:** 3
**Contribution:** 3
**Rating:** 7
**Confidence:** 3

**Summary:**

The present work pertains to analysing and improving Graph Neural Networks operating on geometric graphs that possess E(3) (Euclidean) symmetries, i.e. the tasks of interest are invariant/equivariant to rotations, translations and reflections. The paper challenges the assumption that it is sufficient to use E(3)-equivariant features of 1st degree (the term “degree” is related to the dimensionality of the group representation to which the features are equivariant - 1st-degree features are 3-dimensional and are equivariant to 3D rotations/reflections), as done in EGNN, Satorras et al., ICML’21, which works quite well in practice and is computationally efficient. The authors construct a series of counterexample symmetric geometric graphs (i.e. graphs which, when rotated by certain angles, remain intact up to permutations of their vertices) which are shown to be challenging for geometric GNNs. It is shown that, for certain degrees $l$, a GNN can only produce output features equal to zero, therefore showing that the degree is crucial for expressivity and that $l=1$ is insufficient for some cases. To address this limitation, they combine ideas from EGNN and TFN, Thomas et al., arxiv’18: They create a GNN, similar to EGNN, enriched with auxiliary features of higher degree, using spherical harmonics (similar to TFN). To keep the computational complexity low, they produce the messages used for the updates of node features/positions/auxiliary features via scalarisation using inner products, simplifying the messages used in TFN. The expressivity of the method is theoretically analysed, showing the ability to recover angles, and empirically tested, showing improved performance in several synthetic and real-world benchmarks.

**Strengths:**

**Significance**. The studied problem is important for various physical sciences using data-driven methods and finding better trade-offs between performance and computational complexity is key to their progress. The present paper improves this trade-off both theoretically and empirically.

**Novelty**.
  - *Theoretical results*. The underlying theory illustrating the expressivity barriers of low-degree representations (theorems 3.4-3.6) is simple and concise, yet insightful and of independent technical interest (can be used for other types of symmetric objects to find out properties of neural representations and illustrate limitations of modelling choices). Additionally, theorem 4.1. backs the proposed method with theoretical arguments, hinting that is probably a good trade-off between expressivity and computational complexity (although the former is not completely characterised).
   - *Method*. As far as I know, the proposed approach to incorporate higher-degree representations into message-passing is original. Additionally, it is easy to understand and implement, which may facilitate reproducibility.

**Presentation**. Apart from a few exceptions (see weaknesses), the paper is well-explained, with carefully chosen notation and progressive illustration of the different results (from the theoretical motivation to the model improvement and empirical evaluation).

**Empirical results & Execution**.  The experimental outcomes are in agreement with the theory on synthetic data, while the method is shown to generalise better than its competitors in real-world data, with additional computational efficiency advantages. Overall, this work is well-executed: starting from a characterisation of the limitations of existing works, then proposing a remedy (that gets the best of both worlds from two different families of methods) and finally testing it both on the counterexamples, as well as in real-world scenarios, validating the theory.

**Weaknesses:**

**Clarity: limits audience and creates some issues with contextualization to related work**. Although not a major weakness, there are some issues with the clarity of the presentation, which I think should be addressed to improve the exposition of the paper to a wider audience.
  - In the introduction and the related work section, the authors mention several concepts without providing explanations which might make the text hard to follow, especially for readers not experts in geometric GNNs and Euclidean symmetries.
    - For instance: *x-degree steerable representations* - this is a notion at the heart of the paper, so it should probably be intuitively explained early on, e.g. by summarising the formal definitions at the end of section 3.1.
    - *Clebsch-Gordan tensor product* - this concept is not explained in the paper. However, I do not think that the authors should assume familiarity with it. The reader needs to obtain a clear understanding due to the following: the main methodological innovation of the present work is to replace this operation with inner products to improve computational efficiency. I think that the authors should explain in a separate subsection the differences between these two operators and discuss what they sacrifice by not incorporating CG products in their architecture.
-	Examples 3.2. and 3.3. need further elaboration as they are the main motivation behind the proposed methodology. E.g. why do even folds are not symmetric w.r.t. the cyclic group? Perhaps explain what the dihedral group is. Why are regular polygons symmetric w.r.t. the dihedral group?

**Questions:**

- Section 3.1/Section 4: It is unclear to me why the authors introduced the modulation of the spherical harmonics (re-scaling). How does this affect expressivity?
- Section 3.1: Is the representation of O(3) – Eq. (2) – conventional or introduced here for the first time? In the first case, could the authors provide a reference? In the second case, it would be more appropriate to precisely show that this is indeed a representation for completeness.
- As far as I understand, Eq. (3) refers to graph-level features. Perhaps this should be explicitly mentioned to improve readability.
- What information is lost from the scalarization trick of Eq. (6) compared to CG tensor products? To what extent does this limit expressivity?
- How do the authors explain that in certain cases in Table 2, HEGNN does not achieve 100% performance (although it should according to the theory)? Additionally, why is the std so high (15% and 30%)?
- In table 3, it seems that HEGNN with $l\leq 1$ always outperforms EGNN, which is also 1st-degree. How do the authors explain this?
- It is unclear to me how equivariance to reflections is handled by this method. How does the discussion at the end of section 3.1 affects the operators used?
- Table 5 is quite helpful, perhaps consider moving it to the main text.

**Minor**: I have the impression that there were a few typos throughout the text. I suggest performing a proof-reading (examples: L78 typo: university --> universality, L266 type: donated --> denoted).

**Limitations:**

I recommend that the authors devote a separate section to discuss the limitations of their work. Currently, I could locate only a limited discussion in the appendix with regards to not testing in large-scale systems.

I do not foresee any negative societal impact.

---

> ### Author Rebuttal · Authors · 2024-08-07
>
> Thank you for your comments!  We provide the following responses to your concerns:
>
> > **W1: More detailed background knowledge and conceptual explanations are needed to improve clarity.**
>
> Thank you for your valuable suggestions! We will further summarize the formal definitions of steerable representations at the end of Section 3.1, and introduce the concept of CG tensor product in Section 3.1 and compare its difference with Eq.(6) in Section 4.  Furhter eblations will be added in Examples 3.2 and 3.3. We sincerely thank you once again, and will further enhance the clarity of our paper by providing more explanations related to geometric GNNs, Euclidean symmetries, and Examples 3.2 and 3.3.
>
>
>
> > **Q1: The significance of re-scaling on spherical harmonics.**
>
> Thanks for your comments! The re-scaling part corresponds to the radial function, which is widely used in previous papers such as TFN. Here, for convenience, we combine the radial function into the formulation of the spherical harmonics to improve the readability of our paper.
>
>
>
> > **Q2: Origin of O(3) group representation.**
>
> Nice suggestion! This representation method is conventional and can be found in many physics-related literature, such as [a,b]. More general and theoretical content of representation theory can be found in some pure mathematical books, such as [c,d]. We will add the necessary citations around Eq.(2) in the revised version.
>
> [a] Landau, L. D., and E. M. Lifshitz. Quantum Mechanics: Non-Relativistic Theory.
>
> [b] Chen, J. Q., Jialun Ping, and Fan Wang. Group Representation Theory for Physicists.
>
> [c] Weyl, Hermann. The Classical Groups: Their Invariants and Representations.
>
> [d] Fulton, William, and Joe Harris. Representation Theory.
>
>
>
> > **Q3: It is helpful to explicitly point out that Eq. (3) refers to graph-level features.**
>
> Thank you for raising this point, which is very important for improving the readability of our paper. We will clearly mention this in subsequent revisions.
>
>
>
> > **Q4: What information is lost in the scalarization trick compared to the CG tensor product?**
>
> Thanks for this valuable question! Our HEGNN exclusively passes invariant quantities (inner products of high-degree representations) between features $\tilde v^{(l)}$ of different degrees $l$, unlike the expensive CG tensor product used in TFN, which considers all possible interactions between different frequencies. Our model can be seen as a generalization of the scalarization trick in EGNN to high-degree representations. While the scalarization trick might somehow sacrifice model expressivity in theory, it has shown significantly better efficacy and efficiency in practice compared to conventional high-degree models, as demonstrated by EGNN paper and also our experiments here. Additionally, Theorem 4.1 indicates that passing inner products of full degrees is sufficient to recover the information of all angles between each pair of edges, affirming the theoretical expressivity of our HEGNN in characterizing the geometry of the input structure. As suggested, we will add a new paragraph for the comparision between Eq.(6) and CG tensor products in Section 4.
>
>
>
> > **Q5: Explanation of the results in Table 2.**
>
> Thank you for your comments. The experiment setup in Table 2 is designed from the GWL paper [a]. It requries to first embed the input graphs through an equivariant neural network (e.g. EGNN, HEGNN, TFN), and then classify them through a simple classifier.  As for the phenomenon that HEGNN does not achieve 100% in some cases, it is probably because the classifier is not trained well. We observe that accuracy can be improved to 100%, if increasing the number of training epochs to a sufficiently large value (e.g. 2,000).
>
> The high std is due to the settings used in the GWL paper. There are only 10 test groups in total, and each group is to classify two graphs into two classes. The classification accuracy can only be 0%, 50%, and 100% for each group, which might exlain why the std will be relatively large.
>
> [a] Joshi C K, Bodnar C, Mathis S V, et al. On the expressive power of geometric graph neural networks.
>
>
> > **Q6: The reason for the  performance gap between EGNN and HEGNN$\_{l\leq 1}$.**
>
> Very insightful comment! We speculate that why HEGNN with $l\leq 1 $ outperforms EGNN is mainly owing to HEGNN initialization (Eq.(5)), which is equivalent to an additional massage passing layer, and increases the depth of the neural network. To show this, we additionally test the effects of EGNN-4layer, HEGNN-3layer, and HEGNN-4layer on the N-body dataset, and the results are as follows. All HEGNNs only used 1-st  steerable feature. It can be found that the performance of EGNN-4layer is very close to HEGNN-3layer.
>
> **Table S8:** Comparison between EGNN and HEGNN on N-body
> |N-body ($\times10^{-2}$)|5-body|20-body|50-body|100-body|
> |-|------|-|-|-|
> |EGNN-4layer|0.65|1.01|1.00|1.36|
> |HEGNN$\_{l\leq1}$-3layer|0.63|0.98|0.96|1.31|
> |HEGNN$\_{l\leq1}$-4layer|0.52|0.79|0.88|1.13|
>
>
> > **Q7: How to achieve reflection equivariance with HEGNN?**
>
> Thank you for this valuable question. Since the steerable features are initialized by spherical harmonics in Eq.(5), reflection equivariance is associated with the transformation $(-1)^lD^{{l}}(\mathfrak r)$, where $D^{{l}}(\mathfrak r)$ is the rotation representation. In other words, the sign of the steerable feature changes if $l$ is odd, and keeps unchanged if $l$ is even.  Besides, during the message passing in our model, all steerable features only interact with the scalar (i.e. the inner product), without changing their original equivariance. Hence, reflection equivariance is always represented by $(-1)^lD^{{l}}(\mathfrak r)$ throughout our model.
>
>
> > **Q8: It is suggested to move Table. 5 to the main text.**
>
> Thank you for your suggestion. We will include it in the main text in the revised version.
>
>
> > **Minor: Some typos.**
>
> Thank you very much. We will proof-read our paper again and have the typos fixed.

---

> > ### Comment · Reviewer_3KcU · 2024-08-13
> > **Post rebuttal**
> >
> > I thank the authors for their responses and the additional experiments provided. I strongly encourage them to update their manuscript in several parts as per the reviewers' suggestions (e.g. extra explanations to improve clarity, clearer comparison with CG tensor product to illustrate their technical contribution, additional experiments suggested by reviewer vhjv, computational complexity/runtime discussion suggested by reviewer op9c). I maintain my initial score and positive evaluation of this paper and recommend acceptance.

---

> > > ### Author Response · Authors · 2024-08-13
> > >
> > > Thank you for keeping your positive recommendation on our work. We will definitely and gladly revise our manuscript according to your inspiring suggestions and those of the other reviewers.

---

### Official Review · Reviewer_vhjv · 2024-07-11

**Soundness:** 3
**Presentation:** 3
**Contribution:** 2
**Rating:** 6
**Confidence:** 4

**Summary:**

This paper challenges the prevailing notion that high-degree steerable representations are unnecessary in equivariant Graph Neural Networks (GNNs). The authors provide theoretical analysis showing that equivariant GNNs constrained to 1st-degree representations degenerate to zero functions when applied to symmetric structures like k-fold rotations and regular polyhedra. To address this limitation, they propose HEGNN, a high-degree extension of the EGNN model that incorporates higher-degree steerable vectors while maintaining efficiency through a scalarization technique. The authors evaluate HEGNN on symmetric toy datasets, N-body systems, and MD17 molecular dynamics datasets, demonstrating improved performance over existing models.

**Strengths:**

1. The paper presents a clear and well-motivated research question, challenging an existing assumption in the field of equivariant GNNs.
2. The theoretical analysis is thorough and provides valuable insights into the limitations of low-degree representations for symmetric structures.
3. The proposed HEGNN model offers a promising approach to incorporating high-degree representations while maintaining computational efficiency.
4. The experimental results on symmetric toy datasets align well with the theoretical predictions, providing empirical support for the main claims.
5. The paper includes a good discussion of the trade-offs between expressivity and efficiency in equivariant GNN models.

**Weaknesses:**

1. The experimental comparisons on the N-body system are limited and use different splits and variations compared to existing literature. This makes it difficult to directly compare HEGNN's performance to state-of-the-art methods. Including comparisons with more recent baselines (e.g., ClofNet, GCPNet, SaVeNet) and using standardized benchmarks would strengthen the empirical evaluation.
2. The experiments are primarily focused on predicting future positions of particles/atoms. While this is a relevant task, it may not fully demonstrate the necessity or advantages of high-degree representations across a wider range of equivariant GNN applications. Additional experiments on different tasks or domains could provide more comprehensive evidence to support the paper's claims.
3. Some relevant baselines are missing from the comparisons. For example, GMN-L is not included in the MD17 dataset experiments, despite outperforming the proposed method on several targets. Similarly, GMN is missing from the N-body systems experiment. Including these baselines would provide a more complete picture of HEGNN's performance relative to existing methods.
4. The paper could benefit from a more extensive discussion of the computational trade-offs involved in using high-degree representations. While the authors mention that HEGNN is more efficient than traditional high-degree models, a more detailed analysis of the computational costs and scaling behavior would be valuable.

**Questions:**

1. Can the authors conduct additional experiments with different tasks or application domains to further support the claims presented in the paper? This could help demonstrate the broader applicability and necessity of high-degree representations in equivariant GNNs.
2. How does HEGNN compare to more recent state-of-the-art methods on standardized N-body system benchmarks, such as those used in the ClofNet paper and subsequent works?
3. Can the authors provide a more detailed analysis of the computational efficiency of HEGNN compared to other high-degree models and EGNN? This could include training times, memory usage, and scaling behavior with respect to the number of particles and degree of representations.
4. Have the authors explored the performance of HEGNN on tasks beyond position prediction, such as force field prediction or other molecular property predictions? This could help strengthen the argument for the necessity of high-degree representations.

**Limitations:**

The authors acknowledge that their current experiments are mainly limited to testing on small molecules and have not been verified on large-scale molecules or large-scale physical systems. It remains to be verified whether HEGNN is effective on large-scale geometric graph datasets.

---

> ### Author Rebuttal · Authors · 2024-08-07
>
> Thank you for your comments!  We provide the following responses to your concerns:
>
> > **W1 & Q2: HEGNN should be tested on conventional dataset splits and compared with new baselines such as ClofNet.**
>
> Nice suggestion! We have additionally conducted experiments on standard N-body benchmarks (train/valid/test = 3k/2k/2k), and compared our method with more state-of-the-art methods including ClofNet as well as its variant ClofNet-vel [a], MACE [b] and SEGNN [c], thanks to the availability of their open-source code. The results on standard split and our original protocol are reported in Table S5. It is observed that our HEGNN clearly outperforms existing methods in both cases, indicating the general effectiveness of our model.
>
> > **W2 & Q1 & Q4: Is it possible to test the performance of HEGNN in different prediction targets and different application fields to further illustrate its versatility?**
>
> Thanks for your valuable suggestion. We choose the prediction of future atoms as our task, as it is equivariant, indicating that the input and output spaces share the same coordinate system. With this task, we are capable of evaluating if the output of the model can retain the full geometry including orientation information of the input after multi-layer message passing. We understand that additional experiments on different tasks or domains (e.g. force field prediction or molecular property prediction) are helpful, which, yet, are not the main focus of this paper and are better left for future exploration.
>
> > **W3: It is necessary to supplement baselines such as GMN and GMN-L.**
>
> Thanks for the reminder. As suggested, we have further included the results of GMN-L in the experiments on MD17 in Table S6, and added the comparisons with GMN in the N-body experiment in Table S5. Our HEGNN-6 generally outperforms GMN, and achieves comparable performance to GMN-L in most cases. Given that GMN-L requires careful handcrafting of constraints for chemical bonds into the model design, our model's ability to derive promising results without such enhancements supports its competitive performance. These results will be added to the revised paper.
>
> **Table S5:** Results of N-body dataset under two partitions
> |N-body($\times10^{-2}$)|5-body|20-body|50-body|100-body|
> |-|-|-|-|-|
> |train/valid/test=3k/2k/2k|||||
> |EGNN|0.71|1.08|1.16|1.29|
> |ClofNet|0.89|1.79|2.40|2.94|
> |ClofNet-vel|0.84|1.50|2.28|2.67|
> |GMN|0.67|1.21|1.18|2.55|
> |MACE|1.43|1.93|2.20|2.51|
> |SEGNN|1.81|2.67|3.44|Nan|
> |HEGNN$\_{l\leq1}$|0.64|**0.84**|0.92|1.04|
> |HEGNN$\_{l\leq2}$|0.69|0.89|1.13|**0.94**|
> |HEGNN$\_{l\leq3}$|**0.58**|1.04|**0.92**|1.04|
> |HEGNN$\_{l\leq6}$|0.77|1.06|1.02|1.18|
> |train/valid/test=5k/2k/2k|||||
> |GMN|0.52|0.98|1.04|1.21|
> |ClofNet|0.80|1.49|2.28|2.77|
> |ClofNet-vel|0.78|1.45|2.22|2.62|
> |MACE|1.13|1.60|2.41|3.38|
> |SEGNN|1.68|2.63|3.30|Nan|
> |HEGNN$\_{l\leq1}$|0.52|0.79|0.88|1.13|
> |HEGNN$\_{l\leq2}$|**0.47**|**0.78**|0.90|0.97|
> |HEGNN$\_{l\leq3}$|0.48|0.80|**0.84**|0.94|
> |HEGNN$\_{l\leq6}$|0.69|0.86|0.96|**0.86**|
>
> **Table S6:** Results on MD-17 with GMN & GMN-L
> |MD-17|Aspirin|Benzene|Ethanol|Malonaldehyde|Naphthalene|Salicylic|Toluene|Uracil|
> |-|-|-|-|-|-|-|-|-|
> |GMN|10.14±0.03|**48.12±0.40**|4.83±0.01|13.11±0.03|0.40±0.01|0.91±0.01|**10.22±0.08**|0.59±0.01|
> |GMN-L|**9.76±0.11**|54.17±0.69|4.63±0.01|**12.82±0.03**|0.41±0.01|**0.88±0.01**|10.45±0.04|0.59±0.01|
> |HEGNN$\_{l\leq1}$|10.32±0.58|62.53±7.62|4.63±0.01|12.85±0.01|0.38±0.01|0.90±0.05|10.56±0.10|0.56±0.02|
> |HEGNN$\_{l\leq2}$|10.04±0.45|61.8±5.92|4.63±0.01|12.85±0.01|0.39±0.01|0.91±0.06|10.56±0.05|0.55±0.01|
> |HEGNN$\_{l\leq3}$|10.20±0.23|62.82±4.25|4.63±0.01|12.85±0.02|**0.37±0.01**|0.94±0.10|10.55±0.16|**0.52±0.01**|
> |HEGNN$\_{l\leq6}$|9.94±0.07|59.93±5.21|**4.62±0.01**|12.85±0.01|**0.37±0.02**|**0.88±0.02**|10.56±0.33|0.54±0.01|
>
>
> > **W4 & Q3: Comparison of various indicators at runtime between HEGNN and other high-degree steerable models.**
>
> Thanks! The main difference between our HEGNN and traditional high-degree models is that we employ inner products for message exchange between the representations of different degrees, while TFN resorts to CG tensor products to consider all possible interactions between different representations. By denoting the maximum degree as $L$, the complexity of our inner products is equal to $\textstyle\sum_{l=0}^L(2l+1)=(L+1)^2=O(L^2)$, whereas the complexity of CG tensor products is derived as $O(L^6)$ by [a]. The following tables further report the inference times and model sizes of the high-degree models in the 100-body case of the N-body dataset. It is verified that our HEGNN obtains better performance with lower computation cost, compared to TFN, SEGNN, and MACE.
>
> **Table S7:** Parameters and inference times of models
> |**Parameters of Models**|$l\leq1$|$l\leq2$|$l\leq3$|$l\leq6$|
> |------------------------|-----------|-----------|-----------|-----------|
> |EGNN|134.1k|--|--|--|
> |HEGNN|160.3k|160.9k|161.5k|163.2k|
> |TFN|--|9.6M|19.5M|86.6M|
> |SEGNN|228.1k|244.1k|254.9k|288.1k|
> |MACE|14.8M|38.0M|77.0M|342.5M|
> |**Inference Times ($10^{-2} s$) of Models**|$l\leq1$|$l\leq2$|$l\leq3$|$l\leq6$|
> |EGNN|0.57|--|--|--|
> |HEGNN|0.82|0.88|0.91|1.08|
> |TFN|--|3.75|2.66|OOM|
> |SEGNN|1.33|1.70|1.98|22.33|
> |MACE|22.10|125.87|261.11|OOM|
>
> [a] Passaro S, Zitnick C L. Reducing SO (3) convolutions to SO (2) for efficient equivariant GNNs.

---

### Official Review · Reviewer_7wT3 · 2024-07-15

**Soundness:** 3
**Presentation:** 3
**Contribution:** 2
**Rating:** 7
**Confidence:** 3

**Summary:**

This paper studies the necessity of higher-degree features in geometric graph neural networks (i.e. graph neural networks processing data embedded in three-dimensional space), focusing on the ability to recognize data with non-trivial rotational inner-symmetry such as k-folds and regular polygons, and based on the findings, proposes and validates an extension of the E(n)-GNN architecture. Theoretically, the authors first state in Theorem 3.4 that any O(3)-equivariant function on an inner-symmetric graph must produce an output which has an identical inner-symmetry. This immediately puts a restriction on the possible space of outputs an equivariant function can produce, more specifically as the invariant subspace under the action of the inner-symmetry (Theorem 3.5). It follows that certain inner-symmetric inputs and certain choices of output spaces (i.e. degrees $l$) leads to degenerate output space {0} (Theorem 3.6 and Table 1). Consequently, it follows that using a sufficiently high-degree (even) features leads to non-degenerate output spaces of layers, hence be able to encode e.g. orientations of the input. The authors empirically validate the findings on expressive power in Section 5.1, and then demonstrate good performance of the proposed model based on high-degree features and invariant scalarization on n-body for n <= 100 and MD17 datasets.

**Strengths:**

- S1. The paper is well-written and easy to follow.
- S2. The claims on expressive power regarding symmetric inputs in Section 3.3 is correct, and is original as far as I can confirm; the closest related work is [1], and the authors have clarified the differences of their approach which seems technically correct. This claim is verified by the synthetic experiments in Section 5.1.
- S3. The model proposed by the authors based on the theoretical claims is original as far as I am aware, and it seems useful in practice (Section 5.2-5.3), both in terms of performance and efficiency (slightly more costly than EGNN, but still cheaper than other networks involving higher-degree tensors such as TFN; but it is unclear how the approach compares to MACE, see W4 in the Weaknesses section). In particular, the fact that the model performs well on 100-body simulation task is interesting.

[1] Joshi et al. On the Expressive Power of Geometric Graph Neural Networks (2024)

**Weaknesses:**

- W1. While technically different from [1], this paper conveys a similar message: the use of higher-degree tensors in geometric neural networks is necessary to obtain higher expressive powers which is necessary for certain problems. Given this, one may argue that the GWL hierarchy in [1] is more general (as it handles general non-symmetric inputs as well). Furthermore, inputs with rotational inner-symmetry have been investigated in Section 5.2 of [1] to argue in favor of the higher-degree tensor features, precisely using the task of encoding rotations of the input; while the types of inner-symmetries considered in this work is more general (Table 1), the finding is fundamentally not very different, which can be understood as a weakness of the paper.
- W2. It was not entirely clear to me why the capability of encoding rotations of inner-symmetric inputs would be important for practical learning tasks, although the experiments seem to imply so. For example, on n-body systems, I believe it is very unlikely that one would encounter symmetric inputs including the ones given in Table 1, due to symmetry-breaking factors such as numerical imprecision, simulation errors, and noise. I have similar concerns regarding chemical structures (e.g. MD17), as slight perturbations to the atomic coordinates are sufficient to eliminate the inner-symmetries (if they exist) and hence not experience the problems given in Theorem 3.5 and 3.6. This implies a gap between the theoretical arguments on the expressive power given in Section 3, and empirical results given in Sections 5.2 and 5.3.
- W3. While Theorem 4.1 says that the proposed architecture can recover the information of all angles between each pair of edges, it does not directly address the original problem given in Theorem 3.5 and 3.6.
- W4. I am not sure why MACE [2], while also using higher-degree features and already used for experiments in Tables 1 and 2, is not included for comparison in Tables 3 and 4.

[1] Joshi et al. On the Expressive Power of Geometric Graph Neural Networks (2024)

[2] Batatia et al. MACE: Higher Order Equivariant Message Passing Neural Networks for Fast and Accurate Force Fields (2023)

**Questions:**

Please see the Weaknesses section.

**Limitations:**

The authors have discussed the limitations in the checklist, but not in the main text. I encourage the reader to move the discussions to a separate section in the main text or Appendix.

---

> ### Author Rebuttal · Authors · 2024-08-07
>
> Thank you for your comments!  We provide the following responses to your concerns:
>
> > **W1. Differences and connections with GWL.**
>
> Thank you for raising the comparison with the GWL paper [1]. Here, we would like to further highlight the difference between [1] and our paper:
>
> 1. Different Motivations: the GWL paper aims to investigate the expressivity of different geometric GNNs from the perspective of WL test, whereas our paper focuses more on exploring the necessity of involving high-degree representations. The GWL test paper has discussed rotational inner-symmetry, but the analyses are only derived experimentally.  Upon the formal definition of symmetric graphs, we are able to derive rigorous and theoretical results to explain the expressivity degeneration of equivariant GNNs on symmetric graphs in Table 1. Besides rotational inner-symmetry, we have also investigated regular polyhedra, the derivations of which are not trivial and rely heavily on our derived Theorem 3.6.
> 2. Different Evaluation Scopes: while the GWL paper only discusses the phenomenon of rotational inner-symmetry degradation in a single section, we thoroughly explore more general kinds of symmetric graphs in our experiments. In addition, to evaluate the practical effectiveness of our proposed model, we conduct experimental comparisons on the N-body and MD-17 tasks. The performances align with our theoretical findings in Theorem 3.6 and Theorem 4.1, verifying the strong expressive power of our model.
>
> > **W2. Stability of symmetric structures under perturbations.**
>
> Thanks for your question. We agree that it is unlikely to encounter exactly symmetric inputs in practice, but this does not indicate that our theoretical analyses are NOT practically meaningful. Even though symmetry-breaking factors will make the geometric graph deviate from the symmetric state, the deviated graph is still roughly symmetric. In other words, the outputs of equivariant GNNs on the derivated graphs keep close to zero if the degree value is chosen to be those in Table 1 according to Theorem 3.5 and 3.6, which wil still lead to defective performance. To explore this further, we take the tetrahedron as an example and compare the cases of EGNN, HEGNN$\_{l= 3}$, and HEGNN$\_{l\leq 3}$ when adding noise perturbations.
>
> **Table S3:** Resluts under perturbations
> |                   | $\varepsilon=0.01$ | $\varepsilon=0.05$ | $\varepsilon=0.10$ | $\varepsilon=0.50$ |
> | ----------------- | ------------------ | ------------------ | ------------------ | ------------------ |
> | EGNN              | 50.0 ± 0.0         | 45.0 ± 15.0        | 65.0 ± 22.9        | 60.0 ± 20.0        |
> | HEGNN$_{l=3}$     | 100.0 ± 0.0        | 100.0 ± 0.0        | 100.0 ± 0.0        | 100.0 ± 0.0        |
> | HEGNN$_{l\leq 3}$ | 100.0 ± 0.0        | 100.0 ± 0.0        | 100.0 ± 0.0        | 100.0 ± 0.0        |
>
> Here, $\varepsilon$ represents the ratio of noise, and the modulus of the noise obeys $\mathcal{N}(0,\varepsilon\cdot\mathbb{E}[||\vec x-\vec x_c||]\cdot I)$. It can be observed that the performance of EGNN is slightly improved in the presence of noise (from $50$% when $\varepsilon=0.01$ to $60$% when $\varepsilon=0.5$), while HEGNN demonstrates better robustness. The symmetry-breaking factors you mentioned are very interesting, and the results above will be included in our revised paper.
>
> > **W3. The difference between Theorems 3.5, 3.6, and Theorem 4.1.**
>
> We are sorry for any potential confusion. Since our proposed HEGNN has involved high-degree representations, it does directly address the original problem given in Theorem 3.5 (3.6). The purpose of introducing Theorem 4.1 is to further demonstrate the enhanced expressivity of our HEGNN by using the inner products of full degrees. In other words, Theorem 3.5 (3.6), and Theorem 4.1 discuss different aspects of the benefits by our model.
>
> > **W4. The performance of MACE on the N-body dataset and the MD-17 dataset should be provided.**
>
> Thanks for your reminder. We have additionally tested MACE on the N-body dataset. Due to the expensive running cost, we reduce the channel number of MACE from 64 to 8 (but out of memory still occurs in the 100-body case). Please note that MACE also resorts to CG tensor products similar to TFN, and it is thus consistent that its performance is worse than our model.
>
> **Table S4:** Results of MACE and HEGNN on N-body dataset
> | N-body ($\times 10^{-2}$) | 5-body   | 20-body  | 50-body  | 100-body |
> | :--------------------------- | :------- | :------- | :------- | :------- |
> | MACE                         | 1.13     | 1.60     | 2.41     | 3.38     |
> | HEGNN$\_{l\leq1}$| 0.52     | 0.79     | 0.88     | 1.13     |
> | HEGNN$\_{l\leq2}$| **0.47** | **0.78** | 0.90     | 0.97     |
> | HEGNN$\_{l\leq3}$| 0.48     | 0.80     | **0.84** | 0.94     |
> | HEGNN$\_{l\leq6}$| 0.69     | 0.86     | 0.96     | **0.86** |

---

> > ### Comment · Reviewer_7wT3 · 2024-08-13
> >
> > Thank you for the comprehensive response. The fact that GWL has only empirically investigated the inner-symmetric inputs is something I have missed and indeed addresses the original limitation I have raised. The results in Table S3 seem interesting and important; I propose that the authors include it in the main text in the revision, rather than including it in the Appendix. It would have been great if the performance of MACE on MD-17 was also included, but I guess the results are still valid as it involves comparisons to current SOTA (response to Q4 of reviewer op9c). I have adjusted my score accordingly.

---

> > > ### Author Response · Authors · 2024-08-13
> > >
> > > Thank you for your constructive feedback and willingness to raise the score, which greatly inspired us. We will definitely incorporate your insightful suggestions into our revisions.

---

### Official Review · Reviewer_op9c · 2024-07-15

**Soundness:** 4
**Presentation:** 4
**Contribution:** 2
**Rating:** 6
**Confidence:** 4

**Summary:**

The paper studies the benefit of using higher order steerable features in geometric GNNs and theoretically identifies classes of symmetric geometric graphs where methods using only order-1 (or low order) features are guaranteed to fail.
With this in mind, the authors propose a simple and efficient way to integrate higher order features in the EGNN architecture.

**Strengths:**

Clearly written and well motivated paper.

The proposed architecture seems a valid alternative to existing steerable architectures, enabling the use of higher order features with reduced computational complexity.

The paper also includes a few interesting theoretical insights about the failure cases of EGNN.

**Weaknesses:**

While the overall novelty is a bit limited (the message that higher order features are necessary for expressiveness has been theoretically and empirically explored in a number of previous works in the literature), the theoretical derivation of the failure cases and the proposed architecture can be useful contributions to the community.

See questions below.

**Questions:**

Eq 7, 8: do I understand correctly that this kind of model can not pass any information between features $\tilde{v}^{(l)}$ of different orders $(l)$ beyond the invariant quantities used to compute $m_{ij}$? Then, that essentially means that, after the initial features are extracted via the first convolution with spherical harmonics in Eq. 5, *the rest of model* is invariant to independent rotations by O(3) of each feature $\tilde{v}^{(l)}$, right?
This seems to suggest there should be a gap in performance with models like TFN or SEGNN, which instead allow for different frequencies (features of different orders) to interact inside the model. Can you elaborate on this?


Sec 5.2: it seems important to compare with SEGNN [21] since that work was also motivated by the idea of adapting EGNN to support higher order steerable features, if I remember correctly.


Table 3: I would expect the model size to scale quadratically with the maximum frequency $L$ since $\sum_{l=0}^L (2l+1) = (L+1)^2$, but the runtimes in table 3 do not show this pattern. Why is the case?


Table 4: what are the SOTA on these datasets? It would be good to include the performance achieved by some previous works to give an idea of whether these are competitive results or not


Why introducing Theorem 3.5 when Theorem 3.6 seems more general and practical? Also, I think these theorems hold for any compact (not necessarily finite) subgroups by using the Peter-Weyl theorem.


Line 226: is this a more efficient way to implement this fairly sparse operation?

**Limitations:**

Some limitations are addressed in the supplementary material.

For completeness, I think the authors could comment on the possible gap in expressiveness with other steerable methods like TFN and SEGNN which allow for features of different orders to interact (see my first question above).

---

> ### Author Rebuttal · Authors · 2024-08-07
>
> Thank you for your comments!  We provide the following responses to your concerns:
>
> > **Q1: The performance gap between scalarization-based models (e.g. HEGNN) and high-degree steerable models (e.g. TFN, SEGNN).**
>
> It is true that our HEGNN exclusively passes invariant quantities (inner products of high-degree representations) between features $\tilde v^{(l)}$ of different degrees $l$, unlike the expensive CG tensor product used in TFN or SEGNN, which considers all possible interactions between different frequencies. Our model can be seen as a generalization of the scalarization trick in EGNN to high-degree representations. While the scalarization trick might somehow sacrifice model expressivity in theory, it has shown significantly better efficacy and efficiency in practice compared to conventional high-degree models, as demonstrated by EGNN paper and also our experiments here. Additionally, Theorem 4.1 indicates that passing inner products of full degrees is sufficient to recover the information of all angles between each pair of edges, affirming the theoretical expressivity of our HEGNN in characterizing the geometry of the input structure.
>
>
>
> > **Q2: Comparison with SEGNN is important**.
>
> Nice suggestion! We have additionally compared our model with SEGNN in the table below, using the default settings from the public code. It is observed that SEGNN performs significantly worse than our model (in Table S1). We conjecture that the equivariant non-linear functions applied to the CG tensor products in SEGNN make it difficult to converge to a desirable solution during training, resulting in suboptimal performance. We will include these results in the revised version.
>
>
>
> > **Q3: The relationship between runtimes and model degree.**
>
> Thank you for this valuable observation. While the model size does scale quadratically with the maximum frequency $L $, this does not mean that the exact runtimes in Table 3 follow the same pattern. By leveraging PyTorch, which is based on CUDA toolkit, computations for different frequencies can be largely parallelized. As a result, the actual implementation cost is lower than the quadratic complexity.
>
>
> > **Q4: SOTA on N-body and MD-17**.
>
> Thanks for the reminder. For the N-body dataset, we follow the settings in FastEGNN [a], which was recently published and can be regarded as the SOTA model. As shown in the table below, in all cases from 5-body to 100-body, our HEGNN models generally outperform FastEGNN, verifying the effectiveness of our model's design.
>
> For the MD-17 dataset, we apply the evaluation protocol from the GMN paper [b]. We thoroughly checked the methods in the citations of this paper and found that the GMN-L method proposed in [b] generally performs the best, thus consider it the SOTA method. Our HEGNN-6 achieves comparable performance to GMN-L in most cases. Given that GMN-L requires careful handcrafting of constraints for chemical bonds into the model design, our model's ability to derive promising results without such enhancements supports its competitive performance.
>
> [a] Zhang Y, Cen J, Han J, et al. Improving Equivariant Graph Neural Networks on Large Geometric Graphs via Virtual Nodes Learning.
>
> [b] Huang W, Han J, Rong Y, et al. Equivariant graph mechanics networks with constraints.
>
> **Table S1:** Results on N-body
> |N-body($\times10^{-2}$)|**5-body**|**20-body**|**50-body**|**100-body**|
> |-|-|-|-|-|
> |FastEGNN|0.66|0.81|1.03|0.99|
> |SEGNN|1.68|2.63|3.30|Nan|
> |HEGNN≤1|0.52|0.79|0.88|1.13|
> |HEGNN≤2|**0.47**|**0.78**|0.90|0.97|
> |HEGNN≤3|0.48|0.80|**0.84**|0.94|
> |HEGNN≤6|0.69|0.86|0.96|**0.86**|
>
> **Table S2:** Results on MD-17
> |MD-17|Aspirin|Benzene|Ethanol|Malonaldehyde|Naphthalene|Salicylic|Toluene|Uracil|
> |-|-|-|-|-|-|-|-|-|
> |GMN|10.14±0.03|**48.12±0.40**|4.83±0.01|13.11±0.03|0.40±0.01|0.91±0.01|**10.22±0.08**|0.59±0.01|
> |GMN-L|**9.76±0.11**|54.17±0.69|4.63±0.01|**12.82±0.03**|0.41±0.01|**0.88±0.01**|10.45±0.04| 0.59±0.01|
> |HEGNN≤1|10.32±0.58|62.53±7.62|4.63±0.01|12.85±0.01|0.38±0.01|0.90±0.05|10.56±0.10|0.56±0.02|
> |HEGNN≤2|10.04±0.45|61.8±5.92|4.63±0.01|12.85±0.01|0.39±0.01|0.91±0.06|10.56±0.05|0.55±0.01|
> |HEGNN≤3|10.20±0.23|62.82±4.25|4.63±0.01|12.85±0.02|**0.37±0.01**|0.94±0.10|10.55±0.16|**0.52±0.01**|
> |HEGNN≤6|9.94±0.07|59.93±5.21|**4.62±0.01**|12.85±0.01|**0.37±0.02**|**0.88±0.02**|10.56±0.33|0.54±0.01|
>
>
>
> > **Q5: The relationship between Theorems 3.5 and 3.6.**
>
> Thanks for your comment. In fact, Theorem 3.5 addresses the more general case, while Theorem 3.6 is a special case for practical convenience (since $I-0=I$ is full-rank).
>
> For practical purposes, we consider only the finite symmetric group because only geometric graphs with one or two nodes can exhibit infinite symmetric groups, representing single atoms or diatomic molecules in physics [a]. We do not consider these cases due to their trivial topological structure in realistic tasks. Generalization to compact subgroups can be easily achieved by replacing summation with integration, with volume elements chosen to be normalized and invariant to group action.
>
> From a mathematical perspective, Theorems 3.5 and 3.6 essentially calculate the dimension of the invariant space of a group. This can be achieved by calculating the trace of the projection map of the subgroup [b]. Generalization to compact subgroups is more naturally done by introducing the projection map of compact subgroups, as directly found in [b]. However, for readability, we have not delved deeply into representation theory in this paper. In our future work, we may present our results and further findings using the language of representation theory.
>
> [a] Landau, L. D., and E. M. Lifshitz. Quantum Mechanics: Non-Relativistic Theory.
>
> [b] Fulton, William, and Joe Harris. Representation Theory.
>
> > **Q6: Efficiency of operators in Line 226.**
>
> Yes, this is a more efficient way. In this form, we are able to apply e3nn, which is the most commonly used library for processing spherical harmonics.

---

> > ### Comment · Reviewer_op9c · 2024-08-09
> >
> > I thank the authors for the detailed reply.
> >
> > Regarding Q3, the ratio between the models' sizes for $l\leq6$ and $l\leq1$ is larger than 10 (since $(1+1)^2=4$ vs $(6+1)^2=49$). It still seems surprising to me that a model 12 times larger is only 30% slower. Am I missing something? Is it possible that the model's bottleneck is somewhere else? Or, are other hyper-parameters changed in the model across different values of $L$?
> >
> >
> > Regarding Q4 and Table S1, the (N=5 body) performance reported for SEGNN seems very different from the one in Table 1 in the original paper [21] (1.68 vs 0.43), why is that the case? is the dataset different from the one used in [21]?

---

> ### Author Response · Authors · 2024-08-11
>
> We sincerely appreciate your further comments. We provide more explanations to address your concerns below.
>
> > **Q1: The relationship between model time consumption and degree.**
>
> Your question is both valuable and inspiring! We sincerely apologize for our previous unconsidered response. Upon re-evaluating the complexity of our model, we now believe that your insight is correct—the model's bottleneck lies not only in the value of the degree $L$ but also in the number of channels. As illustrated in the following table, for our models with different $L$ values, the number of channels for type-0 features is fixed to 65, while it is no more than 3 for higher-type features. In other words, the computations for type-0 features account for the majority of the overall running cost. This could explain why the runtimes in Table 3 do not scale quadratically with respect to $L$. Additionally, there are other factors that affect the model's time consumption, including the initialization in Eq. (5) and the coefficient calculations in Eqs. (5) and (8).
>
> Thank you once again for your constructive comment! We will definitely include the above explanations around Table 3 in the revised paper.
>
> **Table S9:** Total dimensions used in HEGNN of different degrees
>
> ||HEGNN$\_{l\leq1}$|HEGNN$\_{l\leq2}$|HEGNN$\_{l\leq3}$|HEGNN$\_{l\leq6}$|
> |-|-|-|-|-|
> |Type-0|65|65|65|65|
> |Type-1|3|3|3|3|
> |Type-2|-|1|1|1|
> |Type-3|-|-|1|1|
> |Type-4|-|-|-|1|
> |Type-5|-|-|-|1|
> |Type-6|-|-|-|1|
> |Total Dim|74|79|86|119|
>
> > **Q2: The gap in performance of SEGNN between our results and original paper.**
>
> Yes, the performance discrepancy of SEGNN can be attributed to the use of a different dataset than the one employed in the original SEGNN paper [21]. In our paper, we utilized the dataset constructed by [a] to provide a more thorough evaluation of the compared methods. This dataset includes a wider range of scenarios, spanning from 5 bodies to 100 bodies, beyond the 5-body case studied in [21]. It is important to note that [a] employed different preprocessing and dataset splitting compared to [21], which may explain why SEGNN's performance differs even in the same 5-body scenario as [21].
>
> In addition to the dataset in [21], here we have conducted additional experiments using the SEGNN dataset, following the official code from the SEGNN paper for dataset construction. We then re-evaluate the performance of EGNN, SEGNN, and our proposed HEGNN models across various scenarios, ranging from 5 bodies to 100 bodies. The results are presented in the following table.
>
> **Table S10:** Comparison between EGNN, SEGNN and HEGNN on N-body from [21]
>
> |N-body ($\times 10^{-2}$)|5-body|20-body|50-body|100-body|
> |-|-|-|-|-|
> |EGNN|0.71|1.04|1.15|1.31|
> |SEGNN|**0.50**|6.61|9.34|13.46|
> |HEGNN$\_{l\leq1}$|0.71|0.97|**0.93**|1.22|
> |HEGNN$\_{l\leq2}$|0.65|**0.91**|1.05|**1.14**|
> |HEGNN$\_{l\leq3}$|0.63|0.99|1.05|1.27|
> |HEGNN$\_{l\leq6}$|0.72|1.05|1.11|1.28|
>
> As observed, SEGNN's performance closely aligns with the results reported in its original paper for the 5-body case (0.50 vs. 0.43), supporting the reliability of our implementation. However, SEGNN shows significantly higher losses in scenarios with a larger number of bodies, indicating a decline in performance as task complexity increases. We conjecture that the steerable nonlinear functions used in SEGNN could make the learning process more difficult when the number of bodies increases.
>
> Although our proposed HEGNN model performs worse than SEGNN in the 5-body case, it consistently exhibits lower loss across various configurations, demonstrating its superior ability to handle increasing task complexity.
>
> We appreciate your valuable attention to this detail and will include the above results and analyses to further strengthen the validity of our findings.
>
> [a] Huang W, Han J, Rong Y, et al. Equivariant graph mechanics networks with constraints.

---

> ### Comment · Reviewer_op9c · 2024-08-12
>
> Thanks for the reply, I think that cleared most of my doubts!
> I encourage the authors to include these details and results in the final version of the manuscript.
>
> I still feel like the novelty is limited, but the paper includes some interesting insights and the proposed method seems effective and seems to be thoroughly evaluated. For these reasons, I maintain my positive recommendation.

---

> > ### Author Response · Authors · 2024-08-12
> >
> > Thank you for your positive recognition of our paper. We will definitly add the details and results in the final version of our paper.  Regarding your mention of our limited novelty, we found that we did not explain it clearly in our previous response. Please allow us to add a few points here.
> >
> > - Although there have been some works studying the importance of using high-degree features, we are the first to rigorously explain the expressivity barriers of low-degree representations. Theorem 3.5 and Theorem 3.6  are simple and concise, yet insightful and of independent technical interest, as also pointed out by Reveiwer 3KcU.
> > - To the best of our knowledge, our HEGNN model is the first to exploit high-degree steerable features using the scalarization trick. This approach not only surpasses prior higher-order models in efficiency but also achieves superior results in experimental results. Besides, its easy implementation may facilitate reproducibility.
> > - In Theorem 4.1, we demonstrate that employ inner products of full degree can recover the information of all angles between each pair of edges, thus affirming the theoretical expressivity of our HEGNN in capturing the geometry of the input structure.
> >
> > Once again, thank you very much for your valuable comments and suggestions that help improve our paper.

---

### Author Rebuttal · Authors · 2024-08-07

# General Response

We sincerely thank all reviewers and ACs for their time and efforts on reviewing the paper. We are very glad that the reviewers recognized the problems we studied, the theories we proposed, and the models we built, and their comments really gave us a lot of inspiration and enlightenment.

For the symmetric structure problem we studied and the theory we proposed, the reviewers thought it was well motivated (Reviewer op9c, vhjv), the theoretical analysis was correct (Reviewer 7wT3), thorough (Reviewer vhjv), insightful and of independent technical interest (Reviewer 3KcU). And our proposed HEGNN, a new method of introducing steerable features through scalarization technique, was evaluated as original (Reviewer 7wT3, 3KcU), promising (Reviewer vhjv), enable the use of higher order features with reduced computational complexity (All Reviewers). The whole article is consistent with the theory and experimental results (Reviewer 7wT3, vhjv), and clear and easy to understand (Reviewer 7wT3, 3KcU).

We also appreciate the reviewers for the insightful comments. To address their concerns, we have added additional experiments as follows.

**Table S1** shows the results of FastEGNN, SEGNN and HEGNN on the N-body dataset with different numbers of particles.

**Table S2****/S6** shows the experimental results of HEGNN, GMN & GMN-L on the MD-17 dataset.

**Table S3** compares HEGNN with EGNN on symmetric graphs when adding noise perturbations, in order to anlalyse how the symmetry-breaking factors influence the performance.

**Table S4** shows the results of HEGNN and MACE on the N-body dataset with different numbers of particles.

**Table S5** compares HEGNN with more relevant baselines under two different protocols on the N-body dataset.

**Table S7** reports the inference times and model sizes of high-degree models.

**Table S8** compares the performance between 4-layer EGNN, 3-layer HEGNN$\_{l\leq 1}$, and 4-layer HEGNN$\_{l\leq 1}$ on the N-body dataset, to explain why  HEGNN$\_{l\leq 1}$ is better than EGNN.

---

### Author Response · Authors · 2024-08-14
**Rebuttal Acknowledgment**

Dear Reviewers and Area Chairs,

We would like to extend our heartfelt gratitude for your dedicated efforts, insightful feedback, and constructive suggestions. Your nice comments and thoughtful suggestions are truly commendable.

Through our discussions and the reviewers' responses, we convice we have addressed the major concerns raised by reviewers. Our work received positive comments from all reviewers both before and after the rebuttal, and now there are two reviewers have given clear acceptance recommendations (Reviewer 7wT3 & 3KcU). This outcome has greatly benefited and encouraged us, and we would like to thank all of you for your valuable support!

Our theory strictly proved the necessity of high-degree steerable features, while our novel framework, HEGNN, proposed a promising method based on scalarization-trick to introduce high-degree steerable features with both efficacy and efficiency. We have made our complete code and training details publicly available, ensuring transparency and reproducibility. The rebuttal points raised during this process will be thoughtfully incorporated into the final version.

Once again, we express our sincere appreciation for your time and dedication in reviewing our work. Your insightful input has significantly contributed to the refinement of our manuscript. Thank you!

---

### Decision · Program_Chairs · 2024-09-25

**Decision:**

Accept (poster)

**Comment:**

Based on the positive reviews and constructive discussion, I recommend accepting this paper. The work presents novel theoretical insights into equivariant Graph Neural Networks and proposes an innovative model (HEGNN) that balances expressivity and efficiency. Experimental results support the theoretical claims and show improvements over existing methods. While minor revisions were suggested, the reviewers agreed on the paper's technical soundness and valuable contribution to the field.